# End-to-end Training for Text-to-Image Synthesis using Dual-Text Embeddings

**Yeruru Asrar Ahmed**                                        *asrar@cse.iitm.ac.in*
*Department of Computer Science and Engineering*
*Indian Institute of Technology Madras*

**Anurag Mittal**                                              *amittal@cse.iitm.ac.in*
*Department of Computer Science and Engineering*
*Indian Institute of Technology Madras*

**Reviewed on OpenReview:** *https://openreview.net/forum?id=gJ10knHV5e*

## Abstract

Text-to-Image (T2I) synthesis is a challenging task that requires modeling complex interactions between two modalities (, *i.e.*, text and image). A common framework adopted in recent state-of-the-art approaches to achieving such multimodal interactions is to bootstrap the learning process with pre-trained image-aligned text embeddings trained using contrastive loss. Furthermore, these embeddings are typically trained generically and reused across various synthesis models. In contrast, we explore an approach to learning text embeddings specifically tailored to the T2I synthesis network, trained in an end-to-end fashion. Further, we combine generative and contrastive training and use two embeddings, one optimized to enhance the photo-realism of the generated images, and the other seeking to capture text-to-image alignment. A comprehensive set of experiments on three text-to-image benchmark datasets (Oxford-102, Caltech-UCSD, and MS-COCO) reveal that having two separate embeddings gives better results than using a shared one and that such an approach performs favourably in comparison with methods that use text representations from a pre-trained text encoder trained using a discriminative approach. Finally, we demonstrate that such learned embeddings can be used in other contexts as well, such as text-to-image manipulation.

## 1 Introduction

Visualizing images for any textual statement is elemental to human understanding of the world. Intelligent systems' ability to generate images from the text for human understanding has a wide range of applications such as information sharing, computer-aided design, text-to-image search, and photo editing. Image synthesis from text is a challenging task due to complex interaction and ambiguous association of the text modality with the image modality. For instance, multiple textual descriptions can describe the same image and vice versa. In addition, finer details of the images may not always be well captured in textual descriptions. In this domain, Generative Adversarial Networks (GANs) (Goodfellow et al., 2014) were the go-to method to generate realistic images. Conditional GANs (Mirza & Osindero, 2014; Odena et al., 2017; Miyato & Koyama, 2018) allow us to generate real images semantically coherent with text (Reed et al., 2016c; Dash et al., 2017; Reed et al., 2016b) by conditioning the generation process on global sentence embeddings.

Though GANs have been shown to generate meaningful images, naively generating high-resolution images from text leads to subpar visual results and training instability due to the complex nature of the task. A set of methods attempts to solve this problem by *advancing the visual generation* part of the model. For instance, StackGAN (Zhang et al., 2017b) employs a *hierarchical stage-wise training* of GANs from a low-resolution to a high resolution and conditions the generator at every stage by images generated from the previous stage generator.

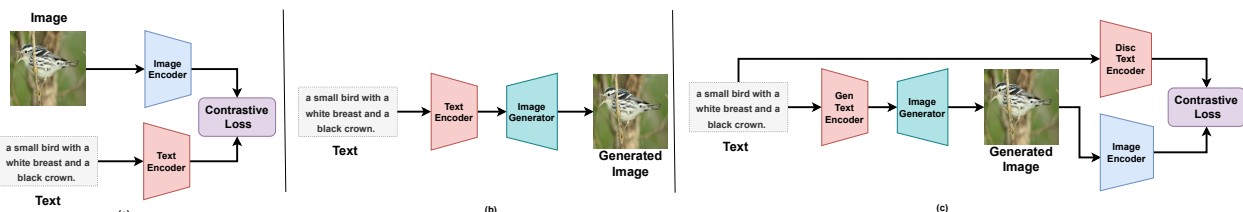

Figure 1:   (a) Text and image are projected into a shared embedding space to enhance mutual information capturing discriminative features (Discriminative Embeddings), (b) Word embeddings are trained by generating images capturing semantic details (Generative Embeddings) and (c) Our Dual Text Embedding approach combining both generative and discriminative embeddings.

Another set of methods (Xu et al., 2018; Zhu et al., 2019; Liang et al., 2020) addresses high-quality image generation from a text by improving the compatibility between text and image modalities. It is achieved by semantically aligning the visual features in image sub-regions with the pre-trained word embeddings through attention. These pre-trained embeddings are trained by projecting text and image features into a shared embedding space and maximising mutual information between text and image features using contrastive loss (Xu et al., 2018; Radford et al., 2021). Furthermore, these embeddings are typically trained generically and reused across various synthesis models (Xu et al., 2018; Liang et al., 2020; Tao et al., 2022; Liao et al., 2022; Ramesh et al., 2021; 2022; Gafni et al., 2022; Rombach et al., 2021). This work explores a new direction for learning text representations within the Text-to-Image (T2I) framework through unified end-to-end training, as illustrated in Figure 1, to enhance compatibility between image and text modalities. Additionally, we combine generative and contrastive training while utilizing two distinct embeddings: one optimized for enhancing the photorealism of generated images and the other focused on capturing accurate text-to-image alignment as shown in Figure 1. We evaluate the proposed approach on three datasets: 1) Oxford-102 (Nilsback & Zisserman, 2008), 2) Caltech-UCSD Birds 200 (CUB) (Welinder et al., 2010), and 3) MS-COCO (Lin et al., 2014b). We use three metrics to assess the generated images: *Inception Score (IS)* (Salimans et al., 2016), *Fréchet Inception Distance (FID)* (Heusel et al., 2017) for image quality, and *R-precision* (Xu et al., 2018) to measure text-image alignment. Our model reduces the FID score from 14.06 to 13.67 on CUB and 40.31 to 30.07 on Oxford-102. For MS-COCO, the FID drops from 35.49 to 25.17, outperforming AttnGAN (Xu et al., 2018), which uses embeddings trained generically in discriminative approach using contrastive loss. Furthermore, we observe that employing separate embeddings gives superior results compared to a shared embedding approach as verified in Section 4.3.4. Finally, we demonstrate that such learnt dual-text embeddings can be used in other contexts as well, such as for text-to-image manipulation.

The contributions of this paper are summarized as follows:

- We propose a novel approach for learning text embeddings specifically tailored to the requirements of Text-to-Image synthesis networks, optimized through an end-to-end training paradigm to enhance synthesis performance.

- We introduce a dual-embedding framework that integrates generative and contrastive training paradigms, with one embedding optimized for photo-realism and the other for robust text-to-image alignment.

- Our approach demonstrates competitive performance compared to methods that utilize text representations derived from pre-trained text encoders optimized using a discriminative training paradigm.

- We show the application of such learnt embeddings for additional tasks, such as text-to-image manipulation.

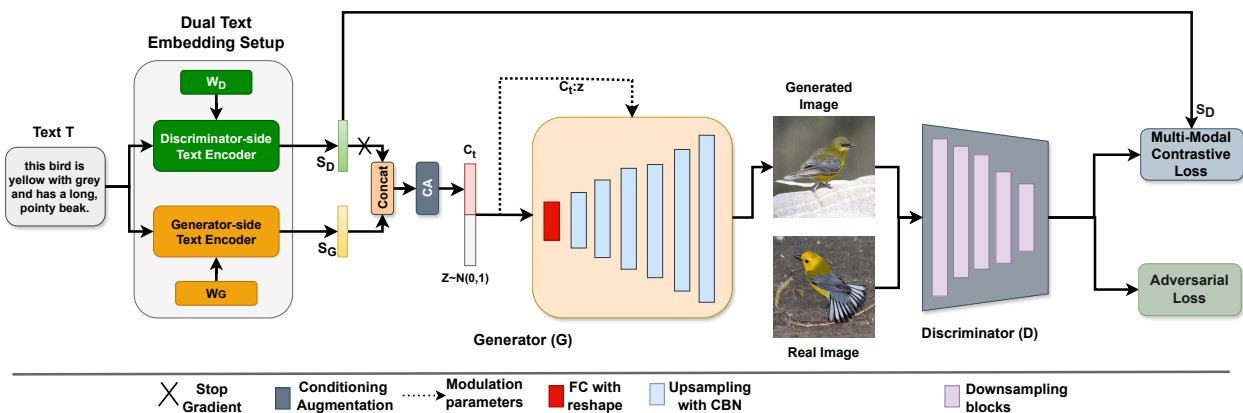

Figure 2: Overview of DTE-GAN architecture. DTE-GAN consists of three core components: i) a single-stage generator $G$ (Section 3.2), ii) a discriminator $D$ (Section 3.3), and iii) a dual text embedding setup (Section 3.4). In the Figure, $W_G$ = generator-side word embeddings, $S_G$ = generator-side sentence embedding, $W_D$ = discriminator-side word embeddings, $S_D$ = discriminator-side sentence embedding. The model is optimised using two objective functions: 1) adversarial loss, and 2) multi-modal contrastive loss.

## 2 Related Work

In this section, some of the relevant works in the literature relating to this paper are discussed briefly.

**Generative Adversarial Networks:** In past few years, GANs (Goodfellow et al., 2014) had been the go-to method for generating images and class-specific images (Mirza & Osindero, 2014; Odena et al., 2017; Miyato & Koyama, 2018) on small datasets such as MNIST (LeCun & Cortes, 2010) and CIFAR (Krizhevsky et al.). However, GAN training is highly unstable when used to generate images on large datasets such as ImageNet (Deng et al., 2009). Researchers have explored to fix this training instability by re-framing GAN loss and regularisation (Arjovsky et al., 2017; Gulrajani et al., 2017; Mao et al., 2017; Miyato et al., 2018; Brock et al., 2017) to generate high-resolution images on large datasets (Karras et al., 2018; Brock et al., 2019).

**Text-to-Image synthesis:** GANs conditioned on global sentence-level embeddings are known to generate meaningful images at low resolutions (Reed et al., 2016b; Dash et al., 2017; Reed et al., 2016c). StackGAN (Zhang et al., 2017b) generates high-resolution images in stage-wise approach, where the generator at each stage is conditioned by the image generated from the previous stage. Unlike StackGAN, HDGAN (Zhang et al., 2018) trains a single generator and multiple discriminators for each resolution. AttnGAN (Xu et al., 2018) uses text embeddings to fine-tune image features and also introduces a multimodal contrastive loss (DAMSM loss) to bridge the gap between generated images and words. DM-GAN (Zhu et al., 2019) refines words and image features using a memory module. MirrorGAN (Qiao et al., 2019b) generates a caption for the generated images that improves the text *vs.* image semantic consistency. SD-GAN (Yin et al., 2019) introduces a Siamese structure for the generator that uses Conditional Batch Normalization (CBN) (Chen et al., 2019) to improve the alignment of text-image. CPGAN (Liang et al., 2020) learns a memory-attended text encoder by attending to salient features in images for each word and fine-grained discriminator (Li et al., 2019). DTGAN (Zhang & Schomaker, 2020) applies channel and spatial attention, conditioned on sentence vector to focus on important features for each textual representation. XMC-GAN (Zhang et al., 2021) maximises the mutual information between text and image using intra-modality and inter-modality contrastive losses. DF-GAN (Tao et al., 2022) uses deep affine transformed global sentence embedding to condition the geand theator and the matching-aware discriminator. In this space of text-to-image semantic alignment-based methods, pre-trained embeddings are an inherent prerequisite. These embeddings are only trained by the discriminative approach. Unlike these methods, our apporach (DTE) attempts to learn text embeddings that capture generative and discriminative properties.

**Generative embedding learning:** Some methods attempt to learn embeddings (text / visual) end-to-end as part of a generator. For example, (van den Oord et al., 2017; Esser et al., 2020) learn discrete embeddings for visual representation and show substantial improvement in the performance of text-to-image synthesis (Ramesh et al., 2021; Ding et al., 2021). Further better and compact representation are learned to improve the quality of image generation (Razavi et al., 2019; Lee et al., 2022; Gafni et al., 2022). Unlike these works that consider only the generation process while learning embeddings, DTE explores the capture of different perspectives of generation and discrimination process by learning dual text embeddings.

**Large Scale Text-to-Image Synthesis:** Denoising Diffusion Probabilistic models (Sohl-Dickstein et al., 2015) have currently achieved remarkable success in image generation (Ho et al., 2020; Nichol & Dhariwal, 2021; Dhariwal & Nichol, 2021) by reversing the *forward Markovian process* with noise removal in multiple steps. Though diffusion-based models are able to generate images with complex and varied interactions for text and generate high-quality images (Ramesh et al., 2022; Saharia et al., 2022; Gu et al., 2022), these approaches require large-scale training and exploit pre-trained discriminative language models like CLIP (Radford et al., 2021). Further, CLIP-based language and image encoders are used as a bootstrapping approach for predicting conditional representations in large-scale GAN-based approaches (Tao et al., 2023; Kang et al., 2023; Zhou et al., 2021). Unlike CLIP, which is trained to capture text-image alignment only, DTE is proposed to learn text-image alignment and text representation toimprove image realism.

## 3 Methodology

In this section, an end-to-end framework called "*Dual Text Embedding GAN*" (DTE-GAN) is formulated for learning text embeddings learn tailored to the T2I synthesis network in an end-to-end manner while capturing different representation for text for improving photo-realism and capturing text-image alignment. In the following sub-sections, the overall architecture is introduced followed by specific details on generator and discriminator architectures respectively; then the final sub-section focuses on learning dual text embeddings.

### 3.1 Model overview

DTE-GAN consists of three core components: 1) a novel dual text embedding setup, 2) a single-stage generator $G$, and 3) a discriminator $D$. The overall architecture is shown in Figure 2.

First, the given text $T$ is passed through the novel dual text embedding procedure to encode the text into two types of sentence embeddings, namely: 1) Generator-side sentence embedding $S_G$, and 2) Discriminator-side sentence embedding $S_D$. The dual text embedding setup consists of two separate Bi-LSTM (Schuster & Paliwal, 1997) text encoders (*Generator-side* and *Discriminator-side*) and their own independent word embeddings ($W_G$ & $W_D$). As the name suggests, the *Generator-side* word embeddings $W_G$ and its encoder are intended to be trained from the image-generation process (generator $G$) and its losses, while the *Discriminator-side* word embeddings $W_D$ and its encoder are optimised by the contrastive loss at the discriminator $D$. Such a decoupling between these two parts of embedding adds flexibility in capturing different natures of image generations and discrimination processes. Specifically, the image generation process warrants learning representation for creating an image, while the discrimination process strives to learn features for improving text-image alignment. Further, the separation of these two embeddings allows $W_G$ to learn from noisy gradients of $G$ independently (as $G$'s gradients are initially noisy due to fake image - sentence pairs), while $W_D$ learns from stable gradients of $D$ (real image - sentence pairs).

During training, $W_G$, the generator-side text encoder, and Generator $G$ are trained from gradient signals of the generation process *i.e.,* Adversarial loss of fake images ($I_{fake}$) and multi-modal contrastive loss between the generated image $I_{fake}$ and the given text $T$. Next, $W_D$ and the discriminator-side text encoder are trained from the gradient signals of the multi-modal contrastive loss between the real image $I_{real}$ and the given text $T$. Further, discriminator $D$ is trained from Adversarial loss ($I_{fake}$, $I_{real}$) and multi-modal contrastive loss between the real image $I_{real}$ and the given text $T$.

### 3.2 Generator

As opposed to other methods that use stacks of GANs or hierarchical GAN, a single-stage generator is employed that can generate an image at any resolution owing to its simpler design and easier training procedure. The generator $G$ takes three inputs: i) a noise vector $z$ of dimension $d_z$ from a Standard Gaussian Distribution $\mathcal{N}(0,1)$ with the Truncation Trick (Brock et al., 2019; Tao et al., 2022; Zhang & Schomaker, 2020), ii) the generator-side sentence embedding $S_G$ of dimension $d_{SG}$, and iii) the discriminator-side sentence embedding $S_D$ of dimension $d_{SD}$. Next, the two sentence embeddings ($S_G$, $S_D$) together are passed through the conditioning augmentation (Zhang et al., 2017b) to get the conditional vector $C_t$ which is concatenated with a noise vector $z$ sampled from a Standard Gaussian Distribution $\mathcal{N}(0,1)$ to form the input vector $f_G$ and passed through a fully connected layer with reshape to create a low-resolution spatial feature map. It may be noted that, in this step, $S_D$ is detached from the gradient flow of $G$ to avoid getting gradients from the generation process (refer to ablation studies in Section 4.3.4).

Further, this low-resolution feature map is passed through a set of upsampling blocks (*UpBlock*) followed by a convolution layer that accepts the last high-resolution feature map and outputs the generated image $I_{fake}$ of dimension $3 \times h \times w$ ($h$ = height, $w$ = width). Each *UpBlock* is formulated as a residual layer consisting of a bi-linear upsampling step followed by two convolution blocks (convolutional layer + Conditional Batch normalisation (CBN)(Chen et al., 2019) + LeakyReLU (Maas et al., 2013)). To increase the stochastic capability of the model, scaled noise is added (similar to StyleGAN (Karras et al., 2019)) to input before passing to the convolutional layer. The modulation parameters in Conditional Batch normalisation (CBN) (Yin et al., 2019; Chen et al., 2019) $\gamma_c$, and $\beta_c$ are calculated from $f_G$ by means of a linear projection layer. The modulation parameters $\gamma_c$, and $\beta_c$ in CBN are calculated as follows:

$$\text{BN}(x \mid C_t, z) = (\gamma + \gamma_c) \cdot \frac{x - \mu(x)}{\sigma(x)} + (\beta + \beta_c) \tag{1}$$

$$f_G = \text{Concat}[C_t, z] \tag{2}$$

$$\gamma_c = FC_\gamma(f_G) \tag{3}$$

$$\beta_c = FC_\beta(f_G) \tag{4}$$

The generator is trained to minimise adversarial loss ($\mathcal{L}_{Adv}^G$), and multi-modal contrastive loss ($\mathcal{L}_{cont}^G$). $\mathcal{L}_{cont}^G$ is formulated as a loss between the features from the generated image and the discriminator-side sentence embeddings. Mathematically, the objective functions can be written as follows:

$$\mathcal{L}_{Adv}^G = \mathbb{E}_{\hat{x} \sim p_G}[-D(\hat{x})] \tag{5}$$

$$\mathcal{L}_{cont}^G \left( \hat{f}_{v_i}, S_{D_i} \right) = -\log \frac{\exp \left( Sim \left( \hat{f}_{v_i}, S_{D_i} \right) \right)}{\sum_{j=1}^N \exp \left( Sim \left( \hat{f}_{v_i}, S_{D_j} \right) \right)} \tag{6}$$

$$Sim(f_v, S_D) = \cos \left( f_v, S_D \right) / \tau \tag{7}$$

Here, $Sim(.,.)$ is a score function to calculate the similarity between sentence embeddings and image features, $\cos(u,v) = u^T v / \|u\|\|v\|$ is the Cosine Similarity between features and $\tau$ denotes the temperature hyperparameter, and $\hat{f}_v$ represents visual features extracted by the discriminator for the generated image $I_{fake}$. We use conditioning augmentation (Zhang et al., 2017b) to sample the sentence condition from an independent Gaussian Distribution $\mathcal{N} \left( \mu \left( s_t \right), \Sigma \left( s_t \right) \right)$. The regularisation term from conditioning augmentation ($\mathcal{L}_{CA}$) for combined sentence embeddings($s_t$) is:

$$\mathcal{L}_{CA} = D_{KL} \left( \mathcal{N} \left( \mu \left( s_t \right), \Sigma \left( s_t \right) \right) \| \mathcal{N}(0, I) \right) \tag{8}$$

Here $\mu(s_t)$ and $\Sigma(s_t)$ are mean and diagonal covaraince matrices that are computed as functions of the combined sentence embedding. The regualarisation term is KL Divergence between the Conditioning Gaussian and a Standard Gaussian Distribution. The final loss for generator is defined as:

$$\mathcal{L}_G = \mathcal{L}_{Adv}^G + \lambda_1 \mathcal{L}_{CA} + \lambda_2 \mathcal{L}_{cont}^G \tag{9}$$

### 3.3 Discriminator

The discriminator $D$ is designed to serve two purposes: (1) to be a critic to determine whether the image is real or fake, and (2) to be a feature encoder to extract image features for multi-modal contrastive loss. The given image ($I_{real}$ or $I_{fake}$) is passed through a series of downsampling blocks (*DownBlocks*), until the feature map is of size $8 \times 8$. Next, these $8 \times 8$ dimensional spatial features are passed through two separate branches: one for extracting features for the adversarial loss and the other for computing image features for multi-modal contrastive loss. For the adversarial branch, the input is passed through a DownBlock, ResBlock, and a fully connected layer to predict the logit to represent if the given image is real or fake. The predicted logit is used as input to an Adversarial Hinge loss (Miyato et al., 2018) $\mathcal{L}_{Adv}^D$ as follows:

$$
\begin{aligned}
\mathcal{L}_{Adv}^D = \mathbb{E}_{x \sim p_{\text{data}}} \left[ \max(0, 1 - D(x)) \right] \\
+ \mathbb{E}_{\hat{x} \sim p_G} \left[ \max(0, 1 + D(\hat{x})) \right]
\end{aligned}
\tag{10}
$$

Here, $x$ and $\hat{x}$ are real ($I_{real}$) and generated ($I_{fake}$) images.

In the multi-modal contrastive loss branch, the features are passed through a DownBlock, ResBlock, and a linear projection layer to output visual features $f_v$. The multi-modal contrastive loss $\mathcal{L}_{\text{cont}}^D$ takes as input the real image features $f_{v_i}$ and sentence embeddings $S_{D_i}$ and calculates the contrastive loss to increase the mutual information in text and image as follows:

$$
\mathcal{L}_{\text{cont}}^D \left( f_{v_i}, S_{D_i} \right) = -\log \frac{\exp \left( Sim \left( f_{v_i}, S_{D_i} \right) \right)}{\sum_{j=1}^N \exp \left( Sim \left( f_{v_i}, S_{D_j} \right) \right)}
\tag{11}
$$

$\mathcal{L}_{\text{cont}}^D$ is the contrastive loss between real image - text pairs. The final objective function for the Discriminator is defined as:

$$
\mathcal{L}_D = \mathcal{L}_{Adv}^D + \lambda_3 \mathcal{L}_{\text{cont}}^D
\tag{12}
$$

### 3.4 Dual text embedding learning

Embeddings can be viewed as memory representation learned by reducing a loss. Multiple embeddings, each learned by optimising on different losses, will capture various memory representations for the same word. The goal of the dual text embedding setup is to learn generator-side word embeddings $W_G$ (along with its encoder) to capture complex representation of words to aid improve the photo-realism of the generated images and discriminator-side word embeddings $W_D$ (along with its encoder) to capture distinctive features for words to align text-image associativity. To achieve this, we make sure that $W_G$ receives only the gradients from image-generation process whereas $W_D$ receives gradients from the contrastive loss. Specifically, we formulate the generator-side embedding loss ($\mathcal{L}_{emb}^G$) and the discriminator-side embedding loss ($\mathcal{L}_{emb}^D$) as follows:

$$
\mathcal{L}_{emb}^G = \mathcal{L}_G
\tag{13}
$$

$$
\mathcal{L}_{emb}^D = \lambda_3 \mathcal{L}_{\text{cont}}^D
\tag{14}
$$

Here, $\mathcal{L}_G$ denotes the loss function for the generator $G$, $\mathcal{L}_{\text{cont}}^D$ denotes the multi-modal contrastive loss between real image ($I_{real}$) features and discriminator-side sentence embedding $S_D$ for the given text T.

## 4 Experiments

In this section, datasets and evaluation metrics are introduced for experiments. Futher, proposed DTE is evaluated and compared quantitatively and qualitatively with other methods in the literature. The specific training details and hyper-parameters are mentioned in the supplementary material.

**Datasets:** DTE-GAN is evaluated on three datasets, namely, 1) Caltech-UCSD birds (CUB) (Welinder et al., 2010), 2) Oxford-102 flowers (Nilsback & Zisserman, 2008), and 3) MS COCO (Lin et al., 2014b) datasets. For CUB and Oxford-102 datasets, we have similar setup to StackGAN (Zhang et al., 2017b).

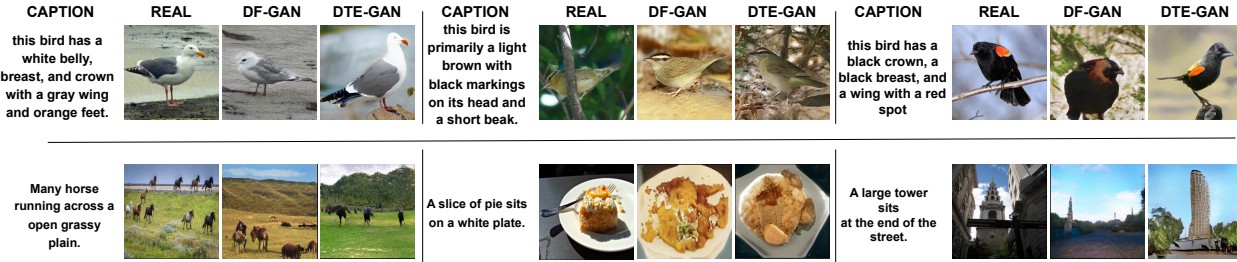

Figure 3: Visual comparision of the images generated by DF-GAN (Tao et al., 2022) and DTE-GAN on CUB(Welinder et al., 2010) and COCO(Lin et al., 2014a) Datasets.

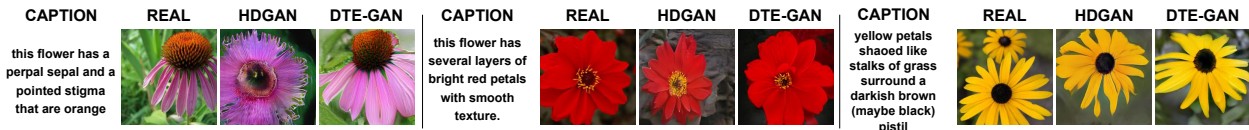

Figure 4: Illustration of the images generated by HDGAN (Zhang et al., 2018) and those of DTE-GAN on Oxford-102 Flower Dataset (Nilsback & Zisserman, 2008).

Ten captions are provided for each image in both the datasets (Reed et al., 2016a). The MS-COCO dataset consists of around 80k training and 40k validation images; and for every image, there are 5 captions provided with the dataset.

| Method | CUB | | | COCO | | |
|---|---|---|---|---|---|---|
| | IS ↑ | FID ↓ | R% ↑ | FID ↓ | R% ↑ | NoP ↓ |
| StackGAN (Zhang et al., 2017b) | $3.70 \pm .04$ | - | - | - | - | - |
| AttnGAN (Xu et al., 2018) | $4.36 \pm .02$ | 23.98 | 67.82 | 35.49 | 83.82 | $230M$ |
| MirrorGAN (Qiao et al., 2019b) | $4.56 \pm .17$ | 18.32 | 57.67 | 34.71 | 74.53 | - |
| DM-GAN (Zhu et al., 2019) | $4.75 \pm .07$ | 16.09 | 72.32 | 32.64 | 88.56 | $46M$ |
| KT-GAN (Tan et al., 2021) | $4.85 \pm .05$ | 17.32 | - | 30.73 | - | - |
| TIME (Liu et al., 2021) | $4.91 \pm .04$ | 14.30 | 71.57 | 31.14 | - | $120M$ |
| DAE-GAN (Ruan et al., 2021) | $4.42 \pm .04$ | 15.19 | **85.45** | 28.12 | **92.61** | $98M$ |
| CSM-GAN (Tan et al., 2022a) | $4.62 \pm .08$ | 20.18 | - | 33.48 | - | - |
| DR-GAN (Tan et al., 2023a) | $4.90 \pm .05$ | 14.96 | - | 27.80 | - | $73M$ |
| ALR-GAN (Tan et al., 2023b) | $4.96 \pm .04$ | 15.14 | 77.54 | 29.04 | 69.20 | $76M$ |
| DF-GAN (Tao et al., 2022) | $4.86 \pm .04$ | 14.81 | - | **19.32** | - | **19M** |
| SSA-GAN (Liao et al., 2022) | **$5.17 \pm .08$** | 15.61 | 85.4 | **19.37** | 90.6 | $26M$ |
| **DTE-GAN** | **$5.12 \pm .04$** | **13.67** | **86.64** | 25.17 | **90.82** | **11M** |
| **DTE-GAN+MAGP** | $5.09 \pm .02$ | **13.94** | 81.33 | **19.69** | 88.39 | **11M** |

Table 1: Quantitative comparison between DTE-GAN and other models on CUB (Welinder et al., 2010) and COCO (Lin et al., 2014a) datasets. "-" indicates values are unreported. The best three results are marked with **bold+underline**, **bold**, and underline, respectively. "↑" indicates the higher, the better, while "↓" indicates the lower, the better.

## 4.1 Visual comparison

The generated images are visually compared between DF-GAN (Tao et al., 2020) and DTE-GAN in the CUB and COCO datasets. In Figure 3, it is evident that DF-GAN struggles to depict complete bird shapes. The presented model, employing dual text embeddings, minimizes image generation loss, improving shape accu-

racy and realistic fine-grained features in generated images. Furthermore, DTE-GAN outperforms DF-GAN in pose representation, resulting in more natural-looking images. With regard to semantic consistency between images and text, DTE-GAN captures detailed structures and overall coherence compared to DF-GAN. Despite not using word embeddings for image region attention, DTE-GAN's ability to learn embeddings for both generation and discrimination allows it to generate images with finer details. DTE-GAN produces images that resemble real images due to its learned embeddings encompassing generation and discrimination aspects. For the COCO dataset, DTE-GAN generates images of comparable quality to DF-GAN while using significantly fewer parameters. This efficiency is achieved by learning tailored text embeddings specifically for the synthesis model. In Figure 4, images for the Oxford-102 dataset are generated and compared with those of HDGAN (Zhang et al., 2018). We can observe that our model is able to capture the complex variation of the flowers and generate more realistic images than HDGAN.

| Method | IS ↑ | FID ↓ |
|---|---|---|
| StackGAN (Zhang et al., 2017b) | $3.20 \pm .01$ | 51.89 |
| StackGAN++ (Zhang et al., 2017a) | $3.26 \pm .01$ | 48.68 |
| HDGAN (Zhang et al., 2018) | $3.45 \pm .07$ | 43.17 |
| SSTIS (Tan et al., 2022b) | $3.37 \pm .05$ | - |
| SS-TiGAN (Tan et al., 2023c) | $3.45 \pm .04$ | 40.54 |
| DualAttn-GAN (Cai et al., 2019) | $4.06 \pm .05$ | 40.31 |
| RAT-GAN (Ye et al., 2024) | 4.09 | - |
| **DTE-GAN** | $\mathbf{4.21 \pm .08}$ | **30.07** |
| **DTE-GAN+MAGP** | **4.26 ± .07** | **31.13** |

Table 2: Quantitative comparison between DTE-GAN and other models on Oxford-102 Dataset. The best three results are marked with **bold+underline**, **bold**, and underline, respectively. "↑" indicates the higher, the better, while "↓" indicates the lower, the better."-" indicates values are unreported.

## 4.2 Quantitative Evaluation

**Evaluation metrics:** To evaluate the quality of images generated, the following metrics used are: *Inception Score (IS)* (Salimans et al., 2016) calculates the Kullback-Leibler (KL) divergence between a conditional distribution and marginal distribution for class probabilities from Inception-v3 (Szegedy et al., 2016) model. Higher IS suggest generated images are from higher diverse classes. *Fréchet Inception Distance (FID)* (Heusel et al., 2017) calculates the Fréchet Distance between two multivariate Gaussians, which are fit to the global features extracted from the Inception-v3 (Szegedy et al., 2016) model on the synthetic and generated images. Lower the FID suggest generated images closer to real images. *R-precision* (R%) precision evaluates text-to-image alignment, by assessing whether generated images can be used to retrieve the text.

The performance of proposed model is compared with that of the lightweight GAN approaches (having similar training setups) for the task of text-to-image synthesis on CUB and COCO datasets in Table 1. From Table 1, on the CUB dataset, we observe that DTE-GAN improves *IS* from 4.91 to 5.12, achieves the best *R-precision* of 86.64 and further decreases *FID* from 14.06 to 13.67. We also train our DTE-GAN with the proposed regularisation trick Matching-Aware Gradient Penalty (MAGP) (Tao et al., 2022), which smooths out discriminator function and allows to generate more realistic images. On the CUB dataset, compared to AttnGAN (Xu et al., 2018) that employs contrastive loss-based embeddings, our model decreases *FID* from 23.98 to 13.67. Such an improvement illustrates the effectiveness of the end-to-end learned dual embeddings over fixed pre-trained embeddings learned on the same data. On MS-COCO (Lin et al., 2014b) dataset (in Table 1), we achieve similar performance of DF-GAN and SSA-GAN with fewer parameters, underscoring the efficiency of the proposed DTE and its ability to learn embeddings tailored specifically to the synthesis model. DTE-GAN+MAGP achieves similar performance as that of SSA-GAN (Liao et al., 2022) on COCO dataset as SSA-GAN and DF-GAN (Tao et al., 2022) both incorporating MAGP.

As shown in the Table 1, we have employed a network with significantly fewer parameters by reducing the width of the layers by half compared to SSA-GAN Liao et al. (2022) and DF-GAN Tao et al. (2020) in their

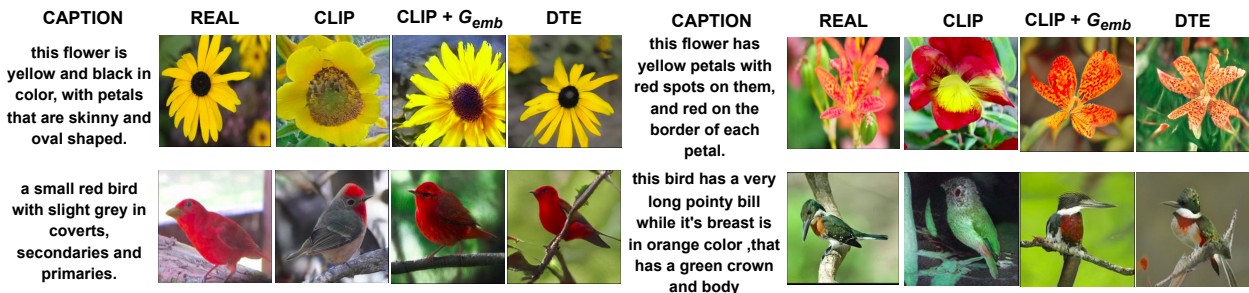

Figure 5: Images generated using CLIP, CLIP + Learnable Generator side embeddings (CLIP + $\mathbf{G_{emb}}$) and DTE on CUB and Oxford-102 datasets.

respective UpBlock and DownBlock. Despite this reduction in complexity, our model achieves comparable results by learning more effective text encoding representations, which improve the performance of Text-to-Image synthesis

Following previous works (Tao et al., 2022; Zhang & Schomaker, 2020; Liao et al., 2022), we report only *FID* scores, as *IS* scores for the MS-COCO dataset do not reflect the quality of the synthesised images. In comparison to TIME (Liu et al., 2021) which learns embeddings along with the Text-to-Image synthesis model, DTE-GAN achieves significant improvement (0.6 in *FID*, +15% in R-precision) demonstrating the effectiveness of dual text embeddings. In Table 2, on the Oxford-102 dataset, we use *IS* and *FID* scores for evaluation, as R-precision scores are not available in the literature. In this dataset, our model improves *IS* score from 4.06 to 4.21 over the state-of-the-art (DualAttn-GAN (Cai et al., 2019), LeicaGAN (Qiao et al., 2019a)) models and remarkably decreases *FID* from 40.31 to 30.07.

## 4.3 Additional Studies

| Dataset | Embeddings | IS ↑ | FID ↓ | R % ↑ |
|---|---|---|---|---|
| | **CLIP** | 4.53 | 21.33 | 74.12 |
| **CUB** | **CLIP+G$_{\mathbf{emb}}$** | 4.51 | 19.36 | 78.35 |
| | **DTE** | **5.12** | **13.67** | **86.64** |
| | **CLIP** | 3.72 | 38.36 | 71.78 |
| **Oxford** | **CLIP+G$_{\mathbf{emb}}$** | 3.81 | 35.93 | 73.87 |
| | **DTE** | **4.21** | **30.07** | **83.19** |

Table 3: We compare quality of T2I generation of our proposed DTE approach with that of models trained using CLIP (Radford et al., 2021) and CLIP+$\mathbf{G_emb}$. "↑" indicates the higher, the better, while "↓" indicates the lower, the better.

### 4.3.1 DTE vs CLIP:

We compare DTE with pre-trained CLIP embeddings by training a GAN for text-to-image generation. Additionally, we train another GAN model resembling the DTE setup. This model uses learnable Generator-side embeddings and pre-trained CLIP embeddings for the discriminator (referred to as CLIP+ $G_{emb}$). Table 3 compares image quality using different CUB and Oxford-102 datasets embeddings. CLIP+$\mathbf{G_{emb}}$ improves over just CLIP. In Figure 5, CLIP images differ from reality, while CLIP+$\mathbf{G_{emb}}$ matches better text and real images due to learned generative embeddings. DTE's combined approach creates images closer to real images. Furthermore, we demonstrate that this approach can be integrated with pre-trained vision-language models, specifically CLIP+$\mathbf{G_{emb}}$ , where only the generator-side embeddings are learned. This method of focusing solely on generator-side embeddings can be seamlessly incorporated into current diffusion-based Text-to-Image models (Rombach et al., 2021; Ramesh et al., 2022) within the existing training frameworks and emphasises learning embedding as needed for synthesis model.

### 4.3.2 Generalisation ability of DTE:

To assess how well the DTE setup applies to other architectures, we integrate it into AttnGAN (Xu et al., 2018), now called AttnGAN+DTE. In AttnGAN, pre-trained text embeddings (DAMSM embeddings) that are generic is used training synthesis model. Instead, we train embeddings scratch using DTE. Incorporating DTE into AttnGAN involves modifying its discriminators to include a dual branch (adversarial loss and multi-modal contrastive loss) after reaching 8×8 feature size. As AttnGAN uses words for alignment in the attention layer, we introduce a word-contrastive loss (Xu et al., 2018; Zhang et al., 2021) in the final discriminator as an additional branch when the feature size is $8 \times 8$, aimed at reducing the semantic gap between words and image features. We combine generator-side and discriminator-side word embeddings to provide word features for the Generator's attention mechanism. The results in Table 4 show that AttnGAN+DTE can train without pre-trained embeddings and even improve the results, proving that DTE can be integrated well with other methods.

| Method | IS ↑ | FID ↓ | R% |
|---|---|---|---|
| AttnGAN | $4.36 \pm .02$ | 23.98 | 67.82 |
| AttnGAN+DTE | $4.38 \pm .03$ | 21.45 | 71.39 |
| DTE-GAN | $\mathbf{5.12 \pm .04}$ | **13.67** | **86.64** |

Table 4: Impact of DTE approach on AttnGAN (Xu et al., 2018) on CUB Dataset (Welinder et al., 2010). "↑" indicates the higher, the better, while "↓" indicates the lower, the better.

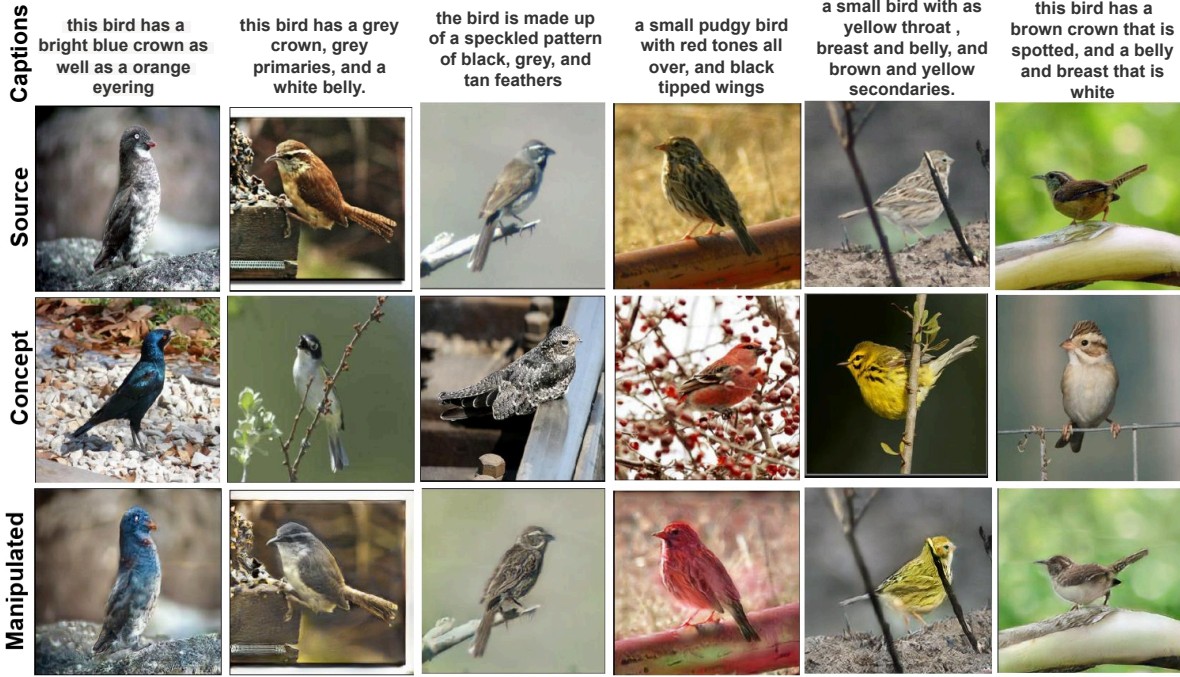

Figure 6: Examples of manipulated images generated by LightWeight GAN (Li et al., 2020b) using DTE-GAN pre-trained embeddings on CUB dataset. Source images are manipulated by the caption of concept images.

### 4.3.3 Application to Text-to-Image manipulation task:

To demonstrate the versatility of the learned dual embeddings, we apply them to text-to-image manipulation tasks. We train dual text embeddings through DTE-GAN on the CUB dataset for text-to-image synthesis. We then utilise the pre-trained word embeddings ($W_G, W_D$) from this synthesis task for text-to-image manipulation in the Lightweight GAN for Text-to-Image manipulations (Li et al., 2020b) task on the same

| Method | IS ↑ | FID ↓ | MP ↑ |
|---|---|---|---|
| MANIGAN | 8.48 | 9.75 | 0.072 |
| LWGAN | 8.26 | 8.02 | - |
| LWGAN w/ DTE-EMB | **8.56** | **7.77** | **0.075** |

Table 5: Quantitative comparison of Inception score, FID and Manipulative metric (MP) (Li et al., 2020a) for manipulated images on CUB dataset. We use Lightweight GAN (Li et al., 2020b) which we name as LWGAN with the pre-trained embeddings using DTE-GAN (LWGAN w/ DTE-EMB). The best results are red. "↑" indicates the higher, the better, while "↓" indicates the lower, the better."-" indicates values are unreported.

dataset. It is important to note that these pre-trained dual text embeddings remain fixed during training of manipulation network. In Table 5, the quantitative performance of the model using these pre-trained embeddings is compared with that of other text-to-image manipulation models. The model improves the Inception score from 8.48 (MANIGAN (Li et al., 2020a)) to 8.56 and reduces the *FID* score from 8.02 (Li et al., 2020b) to 7.77 on the CUB dataset. This demonstrates the ability of dual text embeddings to generalise effectively across different tasks. Figure 6 illustrates few visual examples of the text-to-image manipulations.

| Emb. type | Components | | | | CUB | | | Oxford-102 | | |
|---|---|---|---|---|---|---|---|---|---|---|
| | $\mathcal{L}_G$ | $\mathcal{L}^D_{cont}$ | $S_D \to G$ | $S_G \to D$ | IS ↑ | FID ↓ | R% ↑ | IS ↑ | FID ↓ | R% ↑ |
| Shared | ✗ | ✓ | - | - | $4.54 \pm .04$ | 15.81 | 85.63 | $3.52 \pm .06$ | 33.57 | 81.73 |
| Shared | ✓ | ✓ | - | - | $4.27 \pm .06$ | 18.38 | 82.73 | $3.32 \pm .05$ | 34.83 | 76.15 |
| Dual | ✓ | ✓ | ✗ | ✗ | $4.73 \pm .05$ | 14.93 | 63.79 | $3.84 \pm .03$ | 32.98 | 54.97 |
| Dual | ✓ | ✓ | ✓ | ✓ | $4.25 \pm .05$ | 18.01 | 69.38 | $3.28 \pm .04$ | 35.41 | 70.48 |
| Dual | ✓ | ✓ | ✓ | ✗ | $\mathbf{5.12 \pm .04}$ | **13.67** | **86.64** | $\mathbf{4.21 \pm .08}$ | **30.07** | **83.19** |

Table 6: Quantitative comparison of DTE with its variants. Here, $\mathcal{L}_G$ = generator loss, $\mathcal{L}^D_{cont}$ = multi-modal contrastive loss between real image - text pairs, $S_D \to G$ = whether the generator has access to the discriminator-side sentence embedding, $S_G \to D$ = whether the discriminator has access to the generator-side sentence embedding. "↑" indicates the higher, the better, while "↓" indicates the lower, the better.

### 4.3.4 Importance of dual text embeddings

To verify the effectiveness of the proposed Dual Text Embeddings (DTE) setup, different ways of organising the word embeddings between generator $G$ and discriminator $D$ are evaluated on CUB and Oxford-102 datasets and results shown in Table 6. Specifically, four variants of organising the embeddings are compared, namely: *i*) A shared word embedding layer between $G$ and $D$ that is trained only with a multi-modal contrastive loss $\mathcal{L}^D_{cont}$ between real image-text pairs as it is trained only to capture distinctive features (Table 6, row 1). *ii*) A shared word embedding layer between $G$ and $D$ that is trained using both the generator loss $\mathcal{L}_G$ and real image-text pair contrastive loss $\mathcal{L}^D_{cont}$ to capture distinctive and intricate appearance features in single embeddings (Table 6, row 2). *iii*) A dual embedding setup where $G$ doesn't have access to the discriminator-side sentence embedding $S_D$(Table 6,row 3). *iv*) A dual embedding setup where both $G$ and $D$ have access to the generator-side sentence embeddings $S_G$ and discriminator-side sentence embeddings $S_D$ (Table 6, row 4). The shared embedding model trained only with $\mathcal{L}^D_{cont}$ to capture distinctive features (Table 6, row 1) achieves similar R-precision scores as those of the proposed DTE-GAN, but there is significant drop in *IS* and *FID* scores suggesting that there is drop in the quality of generated images. Next, the shared embedding model trained with both $\mathcal{L}_G$ and $\mathcal{L}^D_{cont}$ performs inferior to the one trained only with $\mathcal{L}^D_{cont}$. It proves that capturing generator-side noisy gradient signals into the same word embeddings degrades the performance. Further, the dual embedding model with independent generator- and discriminator-side embeddings achieves better IS and FID scores compared to those of shared embedding models but has a significant drop in R-precision, suggesting that the images generated are realistic but do not capture text-to-image alignment. Further, allowing the discriminator to have access to the generator-side embeddings significantly drops the performance (Table 6, row 4). As the Discriminator-side embeddings are learned with

| Epoch Count | Components | | CUB | | | Oxford-102 | | |
|---|---|---|---|---|---|---|---|---|
| | $S_D \to G$ | $S_G \to D$ | IS ↑ | FID ↓ | R% ↑ | IS ↑ | FID ↓ | R% ↑ |
| 0 | ✓ | ✓ | $4.21 \pm .04$ | 18.33 | 69.81 | $3.34 \pm .05$ | 36.18 | 71.13 |
| 100 | ✓ | ✓ | $4.59 \pm .06$ | 16.24 | 74.46 | $3.56 \pm .05$ | 33.27 | 75.68 |
| 300 | ✓ | ✓ | $\mathbf{4.82 \pm .04}$ | **14.96** | **78.57** | $\mathbf{3.91 \pm .04}$ | **31.38** | **78.92** |

Table 7: Quantitative comparison of of $(S_G \to D)$ whether the discriminator has access to the generator-side sentence embedding from the epoch count. $S_D \to G$ represents generator has access to the discriminator-side sentence embedding. "↑" indicates the higher, the better, while "↓" indicates the lower, the better.

ground-truth real image-text pairs, it is found beneficial to allow Generator to have access (*i.e.,* a sneak peek) to Discriminator-side embeddings (Table 6, row 5). Discriminator-side embeddings capture distinct features and Generator-side feature captures intricate details to improve photo-realism; providing both the information to Generator allows to generate more realistic and text-aligned images. On the other hand, Generator-side embeddings are learned using noisy gradients from fake images and allowing the discriminator to access them introduces an adverse effect and decreases the performance (Table 6, row 4).

### 4.3.5 Generator-side embeddings to Discriminator

When generator-side embeddings are provided to the discriminator, we observe a significant drop in overall image generation quality, as reported in Table 6. To further evaluate this effect, we have conducted an experiment in which generator-side embeddings are supplied to the discriminator after a specified number of training epochs, with the results presented in Table 7. Specifically, generator-side $S_G$ embeddings are combined with $S_D$ using summation from the start of training (epoch = 0), after 100 epochs when the generator began producing images with plausible structure, and after 300 epochs when more realistic images were generated. The low or noisy quality of generated images during the initial stages affects the learning of generator-side embeddings; consequently, providing these embeddings to the discriminator negatively impacts the overall image generation quality.

| Epoch Count | loss | | CUB | | | Oxford-102 | | |
|---|---|---|---|---|---|---|---|---|
| | $\mathcal{L}_{cont}^D$ | $\mathcal{L}_G$ | IS ↑ | FID ↓ | R% ↑ | IS ↑ | FID ↓ | R% ↑ |
| 0 | ✓ | ✓ | $4.27 \pm .06$ | 18.38 | 82.73 | $3.32 \pm .05$ | 34.83 | 76.15 |
| 100 | ✓ | ✓ | $4.38 \pm .04$ | 16.82 | 83.72 | $3.39 \pm .04$ | 34.04 | 78.69 |
| 300 | ✓ | ✓ | $4.47 \pm .05$ | 16.23 | 84.51 | $3.47 \pm .03$ | 33.96 | 79.94 |
| - | ✓ | ✗ | $\mathbf{4.54 \pm .04}$ | **15.81** | **85.63** | $\mathbf{3.52 \pm .06}$ | **33.57** | **81.73** |

Table 8: Quantitative comparison of shared embeddings trained with contrastive loss ($\mathcal{L}_{cont}^D$) and Generator's loss ($\mathcal{L}_G$) after the epoch count. Single embeddings without $\mathcal{L}_G$ training achieves superior performance. "↑" indicates the higher, the better, while "↓" indicates the lower, the better.

### 4.3.6 Shared Embeddings with Dual Loss

As shown in Table 6, shared embeddings (or single embeddings for text) trained with both contrastive loss $\mathcal{L}_{cont}^D$ and the generator's loss $\mathcal{L}_G$ yield subpar results compared to embeddings trained solely with contrastive loss. We hypothesise that this is because, during the early stages of GAN training, the generator produces predominantly noisy images, negatively impacting the learning of embeddings. To investigate the impact of the generator's loss, we trained multiple networks with shared embeddings and applied the generator's loss to these embeddings at different training stages. The results, reported in Table 8, indicate that applying both losses from the early stages of training adversely affects image generation quality. Moreover, decoupling the embeddings introduces flexibility, enabling them to capture diverse representations, ultimately improving overall image generation quality.

| Epoch Count | CUB | | | Oxford-102 | | |
|---|---|---|---|---|---|---|
| | IS ↑ | FID ↓ | R% ↑ | IS ↑ | FID ↓ | R% ↑ |
| Single | $5.12 \pm .04$ | 13.67 | 86.64 | $4.21 \pm .08$ | 30.07 | 83.19 |
| Dual | $\mathbf{5.14 \pm .04}$ | **13.37** | **87.12** | $\mathbf{4.28 \pm .09}$ | **29.71** | **83.38** |

Table 9: Quantitative comparison of shared embeddings trained with single and dual captions. Single embeddings with out $\mathcal{L}_G$ training achieves superior performance. "↑" indicates the higher, the better, while "↓" indicates the lower, the better.

### 4.3.7 Dual Captions for Dual Text Encoders

Earlier approaches in Text-to-Image synthesis have employed dual (Yin et al., 2019) or multiple captions (Cheng et al., 2020) to enhance semantic consistency and facilitate richer feature extraction. To analyse the impact of using different captions in a dual text encoder setup, we have conducted experiments and reported the results in Table 9. Consistent with previous findings, our results show a marginal improvement in performance. This improvement is attributed to the ability of each text encoder to independently extract superior features, thereby reinforcing the overall semantic alignment and image quality.

## 5 Limitation and Future Scope

This work proposes an approach for learning vision language models through image generation. Due to computational constraints, our model is trained on a single NVIDIA 1080Ti GPU with 12 GB of memory, we limit our experiments to smaller datasets. However, we refer to a comparison with large-scale models in Table 10. As part of future work, we plan to explore vision language learning using diffusion-based models (Ho et al., 2020; Dhariwal & Nichol, 2021) on publicly available large-scale datasets such as WebVid-10M and DiffusionDB (Bain et al., 2021; Wang et al., 2022). Recent advances also highlight the potential of hybrid models that combine GANs with diffusion-based training. In this direction, we intend to investigate the integration of our DTE setup (Discriminator-Teacher Embedding) into large-scale GANs and hybrid models, where the generator is trained using both GANs and diffusion losses (Xu et al., 2024; Sauer et al., 2024). The DTE mechanism can be directly incorporated into such architectures to enhance training efficiency and alignment.

| Model | Type | FID ↓ | NoP ↓ |
|---|---|---|---|
| DALLE (Ramesh et al., 2021) | AutoRegressive | 27.5 | $12B$ |
| LAFITE (Zhou et al., 2021) | GAN | 26.94 | $75M$ |
| Stable Diffusion (Rombach et al., 2021) | Diffusion | 9.62 | $0.9B$ |
| DALLE 2 (Ramesh et al., 2022) | Diffusion | 10.39 | $5.5B$ |
| Make-A-Scene (Gafni et al., 2022) | AutoRegressive | 11.84 | $-$ |
| StyleGAN-T (Sauer et al., 2023) | GAN | 13.90 | $1B$ |
| GigaGAN (Kang et al., 2023) | GAN | 9.09 | $1B$ |
| UFOGen (Xu et al., 2024) | Hybrid | 12.78 | $0.9B$ |
| DTE-GAN **(Ours)** | GAN | 19.69 | $11M$ |

Table 10: Quantitative comparison with large-scale models of FID and Number of Parameters (NoP) on COCO dataset (Lin et al., 2014a). "$-$" indicates values are unreported.

## 6 Conclusion

This study introduces a novel approach to text-to-image synthesis by proposing Dual Text Embeddings (DTE), which learns text embeddings tailored explicitly for the synthesis task in an end-to-end manner. Unlike traditional methods that depend on pre-trained, generic embeddings, DTE uses two separate embeddings designed for specific purposes: one to improve the photo-realism of generated images and the other to

ensure better alignment between text and images. DTE decouples these objectives with dedicated training techniques, leading to enhanced performance.

Our experiments on three benchmark datasets (Oxford-102, Caltech-UCSD, and MS-COCO) demonstrate that having separate embeddings yields better results than using a shared representation. Furthermore, the DTE framework performs favourably compared to methods relying on pre-trained text embeddings optimized with contrastive loss. Additionally, their versatility highlights the adaptability of learned dual embeddings to other language-based vision tasks, such as text-to-image manipulations.

Future work includes extending this dual embedding framework to other multimodal applications, including image or video captioning and visual question answering, further showcasing the potential of task-specific text embeddings in advancing language-vision interactions.

### 6.1 Broader Impact Statement

The proposed Dual Text Embeddings paradigm introduces a new approach where text encoding schemes capture the perspective of current vision-language models and are also trained to capture generators perspective. This approach enables learning text representations that enhance the image synthesis capabilities of conditional generation models and also provides a new paradigm of learning representation by generating images.

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

## A  DTE-GAN with MA-GP

Recent methods (Tao et al., 2022; Zhang & Schomaker, 2020; Liao et al., 2022) have significantly improved quality of synthesised images on COCO dataset (Lin et al., 2014b). Their improved performance may be associated with the Matching-Aware zero-centered Gradient Penalty (MA-GP) term adopted in these methods. MA-GP is applied to Discriminator with real images and its corresponding text to smooth the loss function that allows the Generator to synthesise more realistic images, incorporating MA-GP into DTE-GAN by changing the Discriminator to conditional Discriminator (towards one-way output for gradient penalty). Mismatch pairs are not used as DTE-GAN has a multi-modal contrastive branch for text-image alignment. For conditional prediction, following similar setup of DF-GAN (Tao et al., 2022) of replicating sentence features and concatenating with image features to predict logit values for adversarial loss, the modified adversarial loss function of Discriminator for conditional loss with MA-GP is:

$$
\begin{aligned}
L_D^{Adv} = & - \mathbb{E}_{x \sim \mathbb{P}_{data}}[\max(0, 1 - D(x, S_D))] \\
& + \mathbb{E}_{\hat{x} \sim \mathbb{P}_G}[\max(0, 1 + D(\hat{x}, S_D))] \\
& + k\mathbb{E}_{x \sim \mathbb{P}_{data}}\left[(\|\nabla_x D(x, S_D)\| + \|\nabla_{S_D} D(x, S_D)\|)^p\right]
\end{aligned}
\tag{15}
$$

Here, $k$ and $p$ are hyper-parameters (we use the same hyper-parameter values from DF-GAN (Tao et al., 2022)). For training discriminator-side word embeddings and their sentence encoder from real image-text pairs, we do not update the weights using the gradient of fake conditional prediction from the adversarial loss. Conditional Adversarial loss for Generator is:

$$
L_G^{Adv} = - \mathbb{E}_{\hat{x} \sim \mathbb{P}_G}[D(\hat{x}, S_D)]
\tag{16}
$$

The final objective function for the Generator and Discriminator is defined as:

$$
\mathcal{L}_G = \mathcal{L}_{Adv}^G + \lambda_1 \mathcal{L}_{CA} + \lambda_2 \mathcal{L}_{cont}^G
\tag{17}
$$

$$
\mathcal{L}_D = \mathcal{L}_{GAN}^D + \lambda_3 \mathcal{L}_{cont}^D
\tag{18}
$$

## B  Text Encoding Scheme

We have used a single-stage Bi-LSTM Schuster & Paliwal (1997) for text encoding, following the popular DAMSM Xu et al. (2018) embeddings commonly employed in lightweight GAN models Xu et al. (2018); Zhu et al. (2019); Liao et al. (2022); Tao et al. (2020). The DAMSM embeddings are trained to learn discriminative features by distinguishing between instances, ensuring a fair comparison focused on design principles rather than simply increasing the number of parameters. Additionally, we have conducted an ablation study by replacing the Bi-LSTM (Schuster & Paliwal, 1997) with a 4-layer Transformer encoder Vaswani et al. (2017) in both the generator and discriminator text encoders, and we have reported the results in Table 11.

| Dataset | Encodings | IS ↑ | FID ↓ | R % ↑ |
|---------|-----------|------|-------|-------|
| CUB | **Bi-LSTM** | 5.12 | 13.67 | 86.64 |
|     | **Transformer Encoder** | **5.19** | **13.12** | **87.9** |
| Oxford | **Bi-LSTM** | 4.21 | 30.07 | 83.19 |
|        | **Transformer Encoder** | **4.27** | **29.61** | **83.94** |

Table 11: We compare quality of T2I generation using Bi-LSTM and 4 layer Transformer Encoder text encoding scheme and report the results on CUB and Oxford-102 dataset.

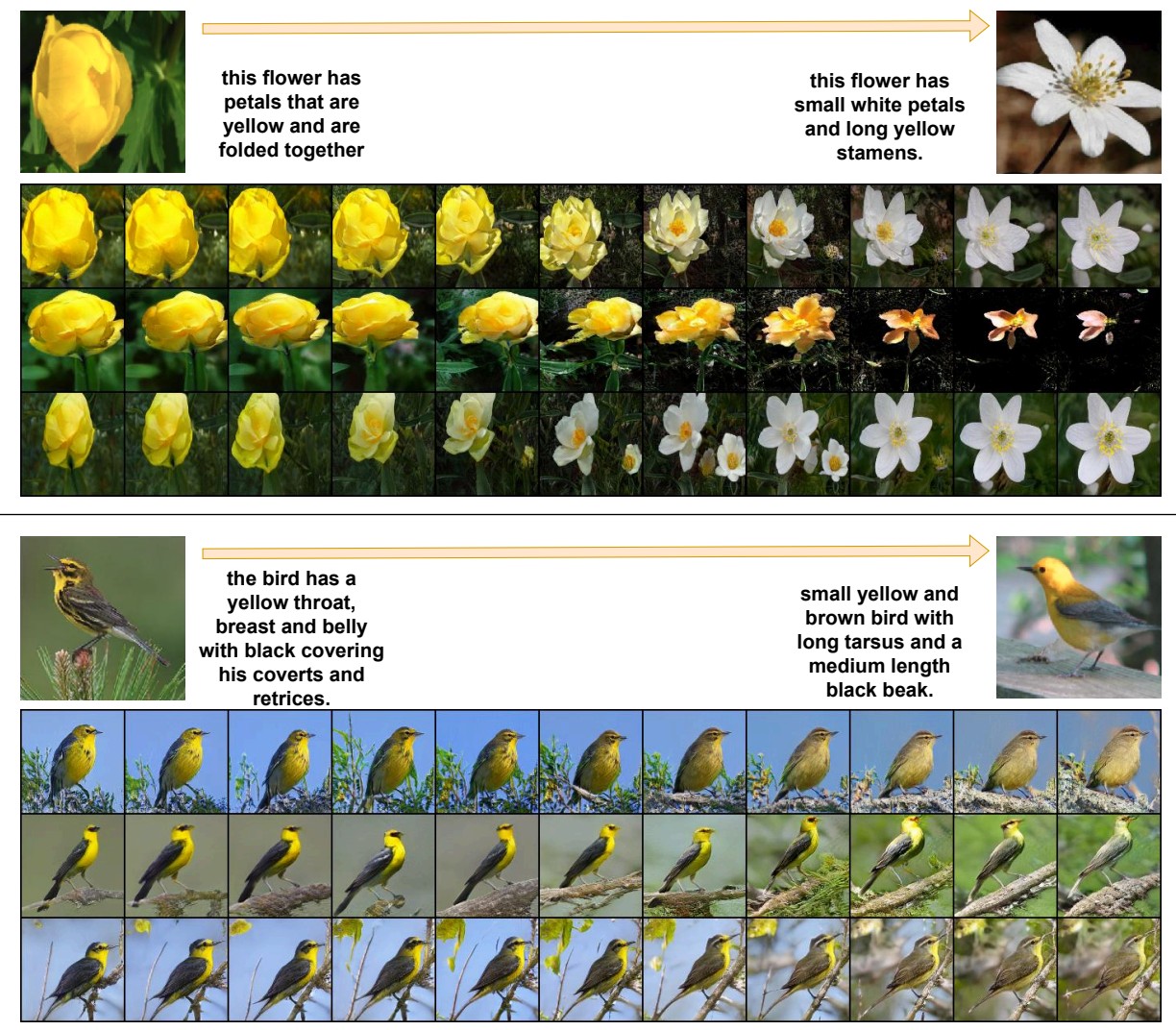

Figure 7: Images generated via sentence interpolation between captions (for reference images), with interpolation applied only to Discriminator-side embeddings $\mathbf{S_D}$ (top row), only to Generator-side embeddings $\mathbf{S_G}$ (middle row) and to both embeddings $\mathbf{S_D}$&$\mathbf{S_G}$ (bottom row) shown for flowers and birds using DTE-GAN.

## C Sentence Interpolation

Figure 7 illustrates sentence interpolation using different sets of embeddings. We perform interpolation on (i) discriminator-side embeddings alone (top row), (ii) generator-side embeddings alone (middle row), and (iii) both embeddings together (bottom row).

We observe that interpolating only the discriminator-side embeddings trained to capture text-image alignment results in images that exhibit smooth transitions in high-level concepts such as shape and object features. In contrast, interpolating only the generator-side embeddings (while keeping the discriminator-side fixed) primarily affects the style and visual appearance, where the first image gradually adopts stylistic elements of the second. Finally, when both generator and discriminator embeddings are interpolated, the generated images show more comprehensive changes in concept and style, producing more coherent, semantically rich outputs and closer to realistic images.

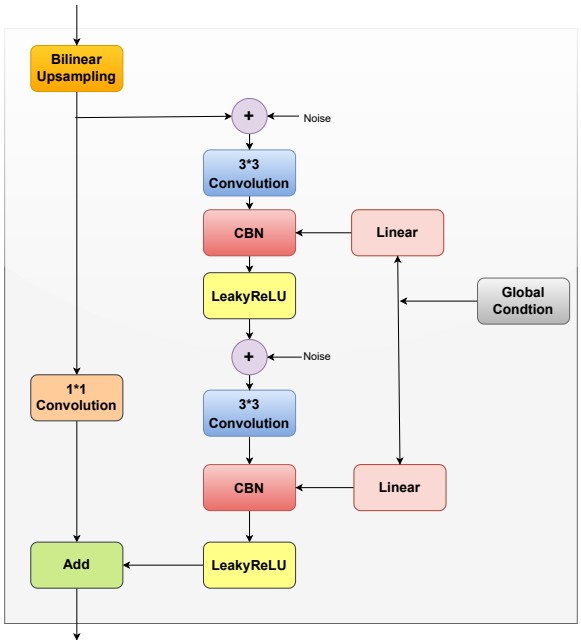

Figure 8: UpBlock used in Generator of DTE-GAN.

For sentence interpolation, we follow the Text-to-Image synthesis approach proposed by Reed et al. (2016c), where an interpolated sentence embedding $S_{INT}$ is computed between two embeddings $S_1$ & $S_2$, derived from different captions, using a weighted summation as follows (with $\alpha \in [0, 1]$):

$$S_{INT} = (1 - \alpha)S_1 + \alpha S_2 \tag{19}$$

# D  Details of the Proposed Architecture

In this section, we elaborate internal architecture details of the DTE-GAN. Proposed model is implemented using Pytorch (Paszke et al., 2019) framework. DTE-GAN architecture consists of a dual text embedding setup (Section D.0.1), a single-stage Generator (Section D.0.2) and a Discriminator (Section D.0.3).

### D.0.1  Dual Text Embeddings

In the Dual Text Embeddings setup, bi-Directional LSTM (Schuster & Paliwal, 1997) are used as text encoder both generator-side and discriminator-side. For each direction in the LSTM, hidden layer size is set as 128. The size of word embeddings $W_D$ and $W_G$ is set to 256. The sentence embeddings $S_G$, $S_D$ are encoded from the output of last hidden state of respective text encoders. For both the text encoders, sentence embedding size is set to 256.

### D.0.2  Generator

Single-stage generator $G$ is used to generate $256 \times 256$ resolution images with base channel dimension of 64. Details of $G$'s architecture are shown in Table 12. The generator $G$ takes noise $z$ along with generator-side sentence embeddings $S_G$ and the discriminator-side sentence embedding $S_D$ and passes them through a set of linear layers followed by a set of upsampling blocks (UpBlocks). UpBlock at each stage is utilised for up sampling spatial features as shown in Figure 8. $S_G$ and $S_D$ are also used to calculate modulation parameters for Conditional Batch Normalisation (Yin et al., 2019). Features are passed through a self modulation convolution and a $1 \times 1$ convolution resulting in generation of final image of dimension $3 \times 256 \times 256$

| |
|---|
| $z\epsilon\mathbb{R}^{100} \sim \mathcal{N}(0, I)$ , $S_G$ $\epsilon$ $\mathbb{R}^{256}$, $S_D$ $\epsilon$ $\mathbb{R}^{256}$, $W_G$ $\epsilon$ $\mathbb{R}^{256}$, $W_D$ $\epsilon$ $\mathbb{R}^{256}$ |
| Linear(512) $\longrightarrow$ 512 |
| Conditional Augmentation(512) $\longrightarrow$ 200 |
| Linear(200+100) $\longrightarrow$ $(8 * ch) \times 4 \times 4$ |
| UpBlock $\longrightarrow$ $(8 * ch) \times 8 \times 8$ |
| UpBlock $\longrightarrow$ $(8 * ch) \times 16 \times 16$ |
| UpBlock $\longrightarrow$ $(4 * ch) \times 32 \times 32$ |
| UpBlock $\longrightarrow$ $(2 * ch) \times 64 \times 64$ |
| UpBlock $\longrightarrow$ $(2 * ch) \times 128 \times 128$ |
| UpBlock $\longrightarrow$ $ch \times 256 \times 256$ |
| Self Modulation Convolution $\longrightarrow$ $ch \times 256 \times 256$ |
| $1 \times 1$ Convolution $\longrightarrow$ $3 \times 256 \times 256$ |

Table 12: Generator architecture of DTE-GAN. Base channel dimension $ch = 64$.

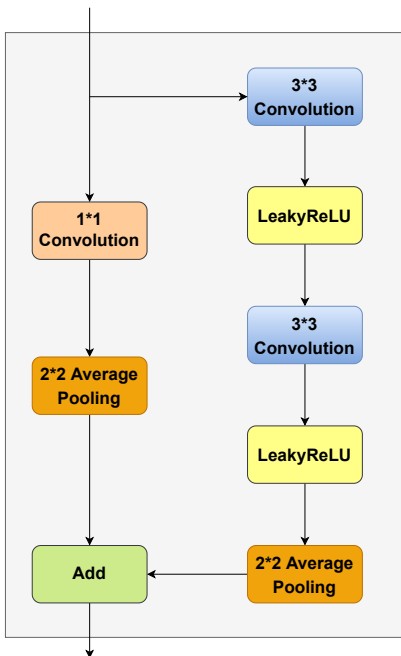

Figure 9: DownBlock used in Discriminator of DTE-GAN.

### D.0.3 Discriminator

Discriminator $D$ is utilised to provide adversarial loss and also act as a feature extractor for multi-modal contrastive loss (as shown in Table 13). Unlike multiple / multi-stage discriminator setup, presented model with a single discriminator, is easy to train and not having a cumbersome training procedure. The discriminator $D$ takes image of dimension $3 \times 256 \times 256$ and passes it through a set of down sampling blocks (DownBlocks- as shown in Figure 9) followed by two branches, one for the adversarial loss and the other multi-modal contrastive loss.

| RGB images $3 \times 256 \times 256$, $S_D \; \epsilon \; \mathbb{R}^{256}$, $W_D \; \epsilon \; \mathbb{R}^{256}$ | |
|:---:|:---:|
| DownBlock $\longrightarrow ch \times 128 \times 128$ | |
| DownBlock $\longrightarrow (2 * ch) \times 64 \times 64$ | |
| DownBlock $\longrightarrow (4 * ch) \times 32 \times 32$ | |
| DownBlock $\longrightarrow (4 * ch) \times 16 \times 16$ | |
| DownBlock $\longrightarrow (4 * ch) \times 8 \times 8$ | |
| DownBlock $\longrightarrow (8 * ch) \times 4 \times 4$ | DownBlock $\longrightarrow (8 * ch) \times 4 \times 4$ |
| ResBlock $\longrightarrow (8 * ch) \times 4 \times 4$ | ResBlock $\longrightarrow (8 * ch) \times 4 \times 4$ |
| Linear$((8 * ch) \times 4 \times 4) \longrightarrow 1$ | Linear$((8 * ch) \times 4 \times 4) \longrightarrow 256$ |
| Adversarial Loss | Multi-Modal Contrastive Loss |

Table 13: Discriminator architecture of DTE-GAN. Base channel dimension $ch = 64$.

## D.1 Implementation Details

Implementation of the models is done using the PyTorch framework (Paszke et al., 2019) and optimising the network using Adam optimiser (Kingma & Ba, 2015) with the following hyper parameters: $\beta_1 = 0.5$, $\beta_2 = 0.999$, batch size = 24, learning rate = 0.0002, $\lambda_1 = 1$, $\lambda_2 = 1$ and $\lambda_3 = 1$. Spectral Normalisation (Miyato et al., 2018) is used for all convolutions and fully connected layers in generator and discriminator. The model is trained for 600 epochs on CUB and Oxford-102 datasets (takes $\sim$4 days in 2 NVIDIA 1080Ti GPUs) and 120 epochs for COCO dataset (takes $\sim$7 days in 2 NVIDIA 1080Ti GPUs). During inference, we report results with exponential moving average weights, with a decay rate of 0.999.

