# Supplymentary: End-to-end Training for Text-to-Image Synthesis using Dual-Text Embeddings

**Yeruru Asrar Ahmed**  *asrar@cse.iitm.ac.in*
*Department of Computer Science and Engineering*
*Indian Institute of Technology Madras*

**Anurag Mittal**  *amittal@cse.iitm.ac.in*
*Department of Computer Science and Engineering*
*Indian Institute of Technology Madras*

**Reviewed on OpenReview:** *https://openreview.net/forum?id=gJ1OknHV5e*

## 1 More Qualitative Results

Additional qualitative results are illustrated of DTE-GAN on CUB Welinder et al. (2010) dataset in Figures 1 and 2, on Oxford-102 Nilsback & Zisserman (2008) dataset in Figures 3 and 4, and on MS-COCO Lin et al. (2014) dataset in Figures 5 and 6 . Comparing these results with those of AttnGANXu et al. (2018) and DM-GANZhu et al. (2019) on CUB dataset, HDGANZhang et al. (2018) on Oxford-102 dataset, and AttnGAN for MS-COCO dataset. Additional qualitative results are provided for image manipulations using LightWeight GANLi et al. (2020) bootstrapped with DTE-GAN pre-trained embeddings in Figures 7 and 8.

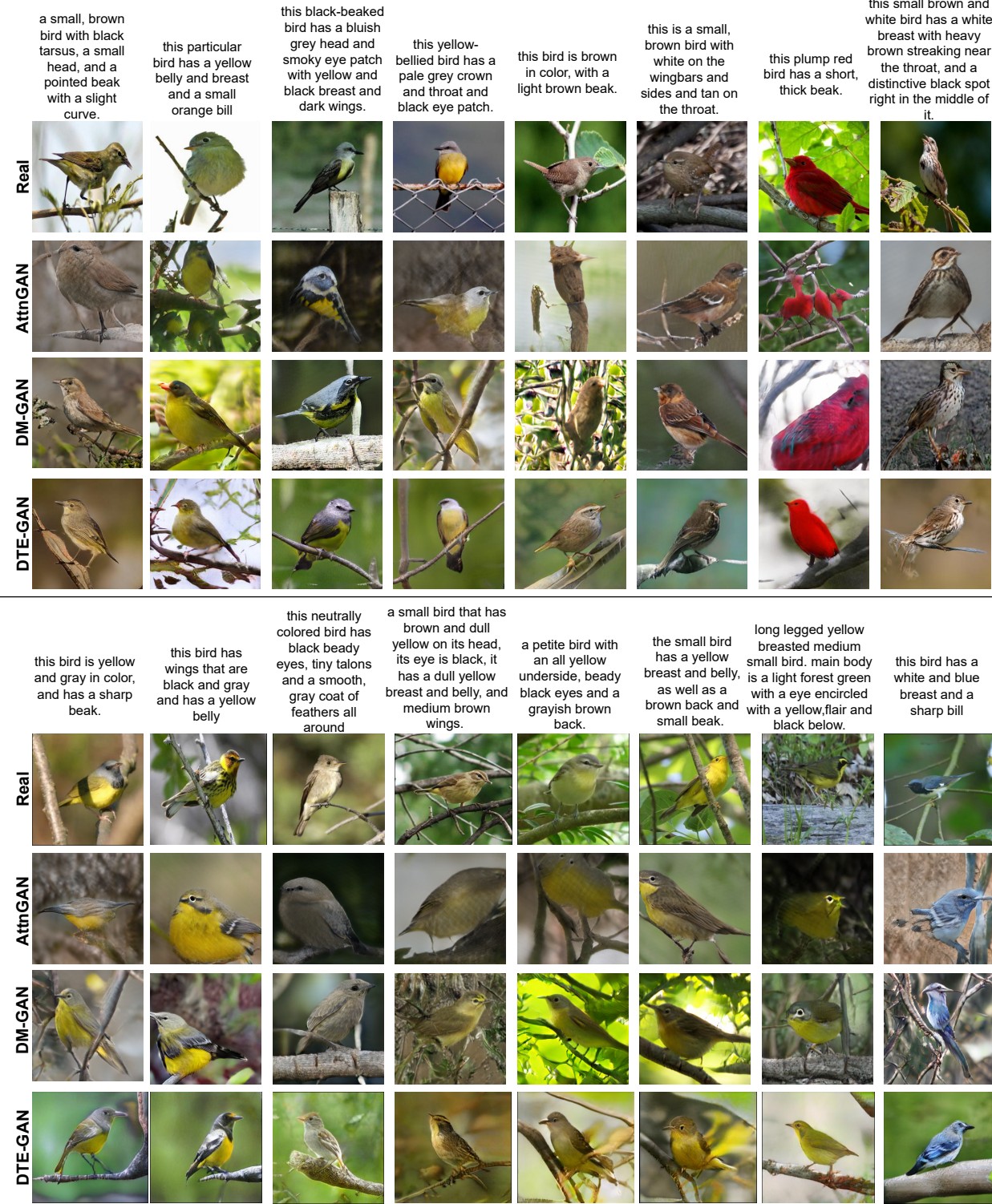

Figure 1: Images generated by DTE-GAN (ours) on CUB dataset and compared with AttnGAN Xu et al. (2018) & DM-GAN Zhu et al. (2019).

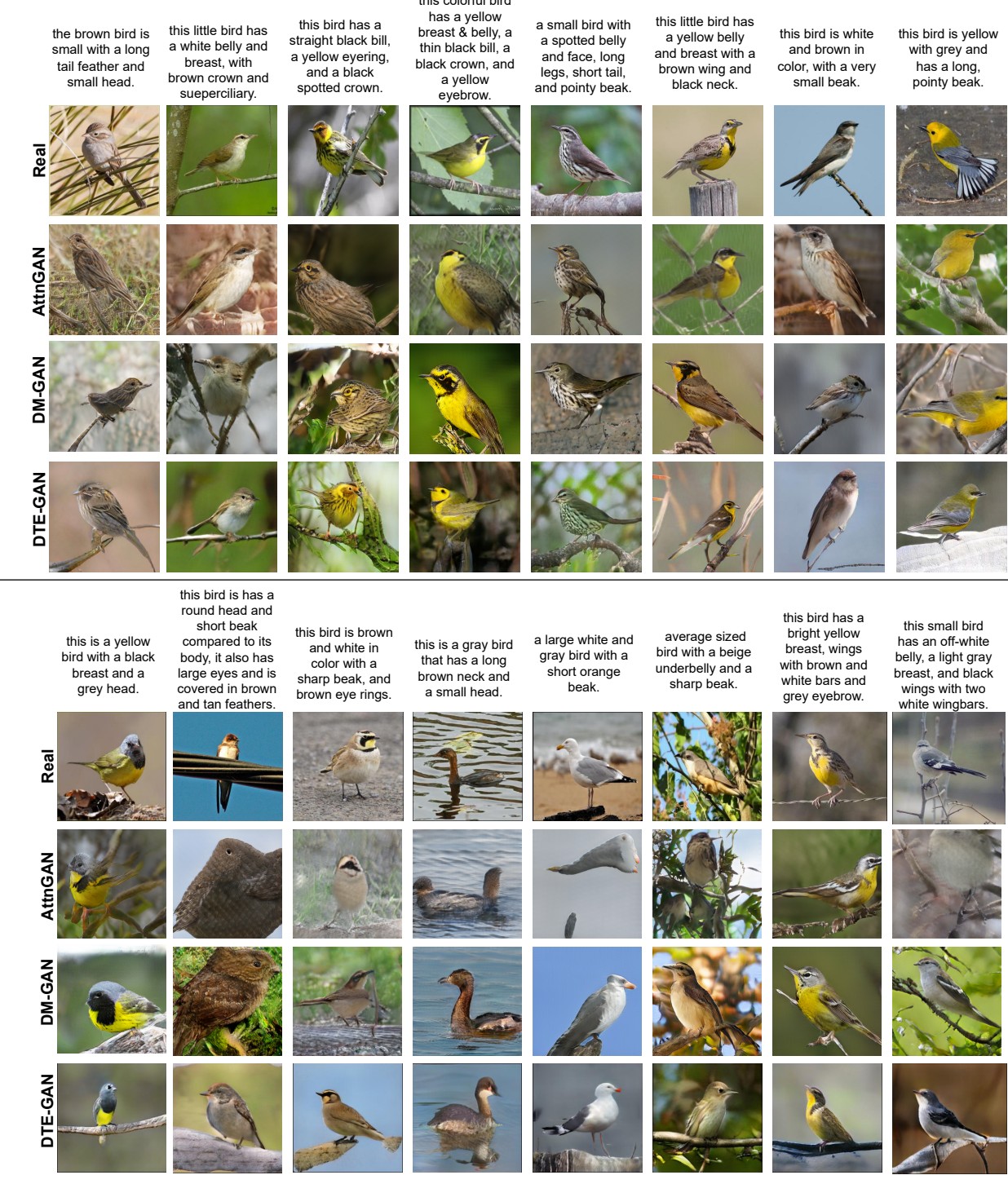

Figure 2: Images generated by DTE-GAN (ours) on CUB dataset and compared with AttnGAN Xu et al. (2018) & DM-GAN Zhu et al. (2019).

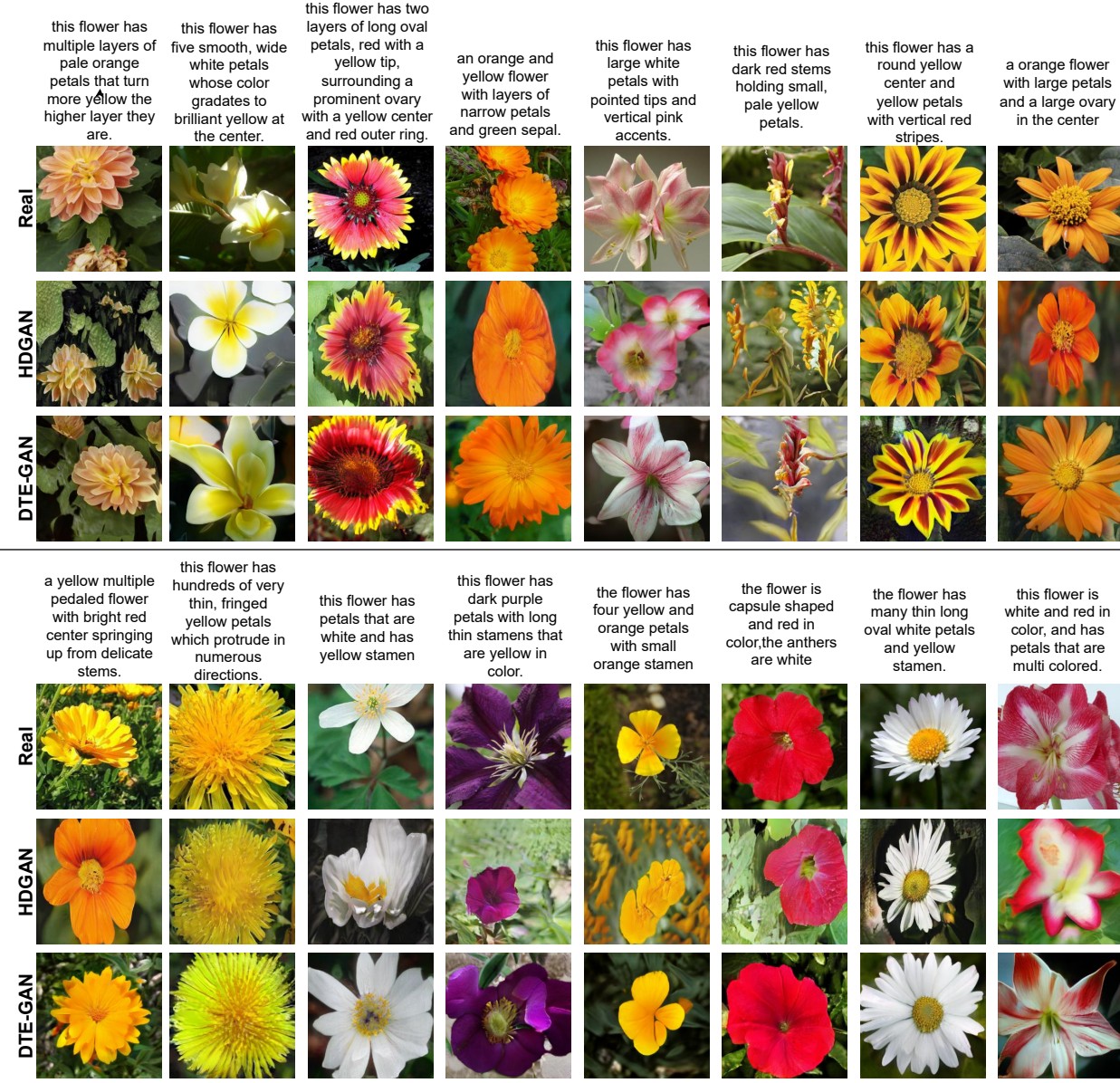

Figure 3: Images generated by DTE-GAN (ours) on Oxford-102 dataset and compared with HDGAN Zhang et al. (2018).

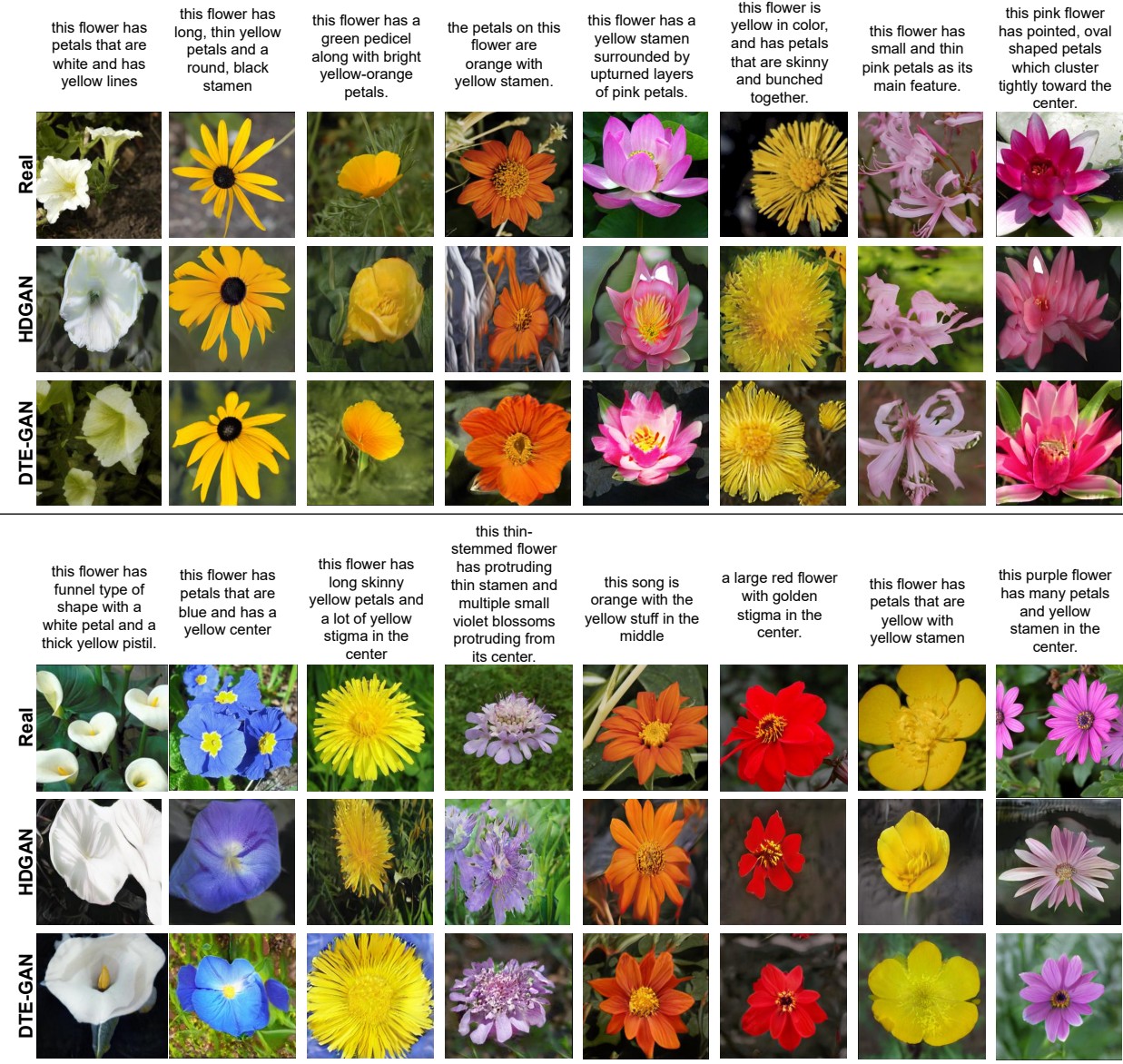

Figure 4: Images generated by DTE-GAN (ours) on Oxford-102 dataset and compared with HDGAN Zhang et al. (2018).

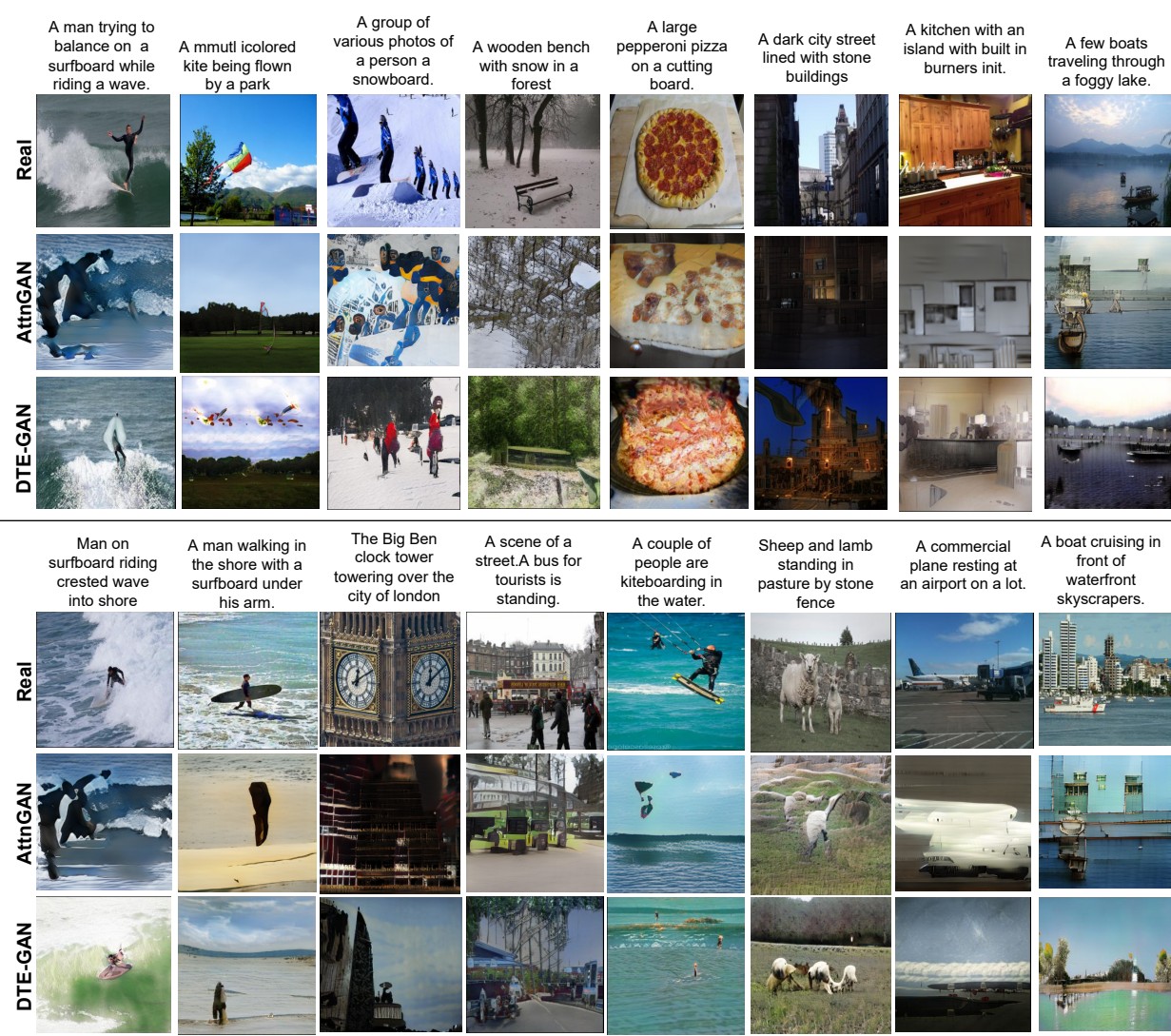

Figure 5: Images generated by DTE-GAN (ours) on COCO dataset and compared with AttnGAN Xu et al. (2018).

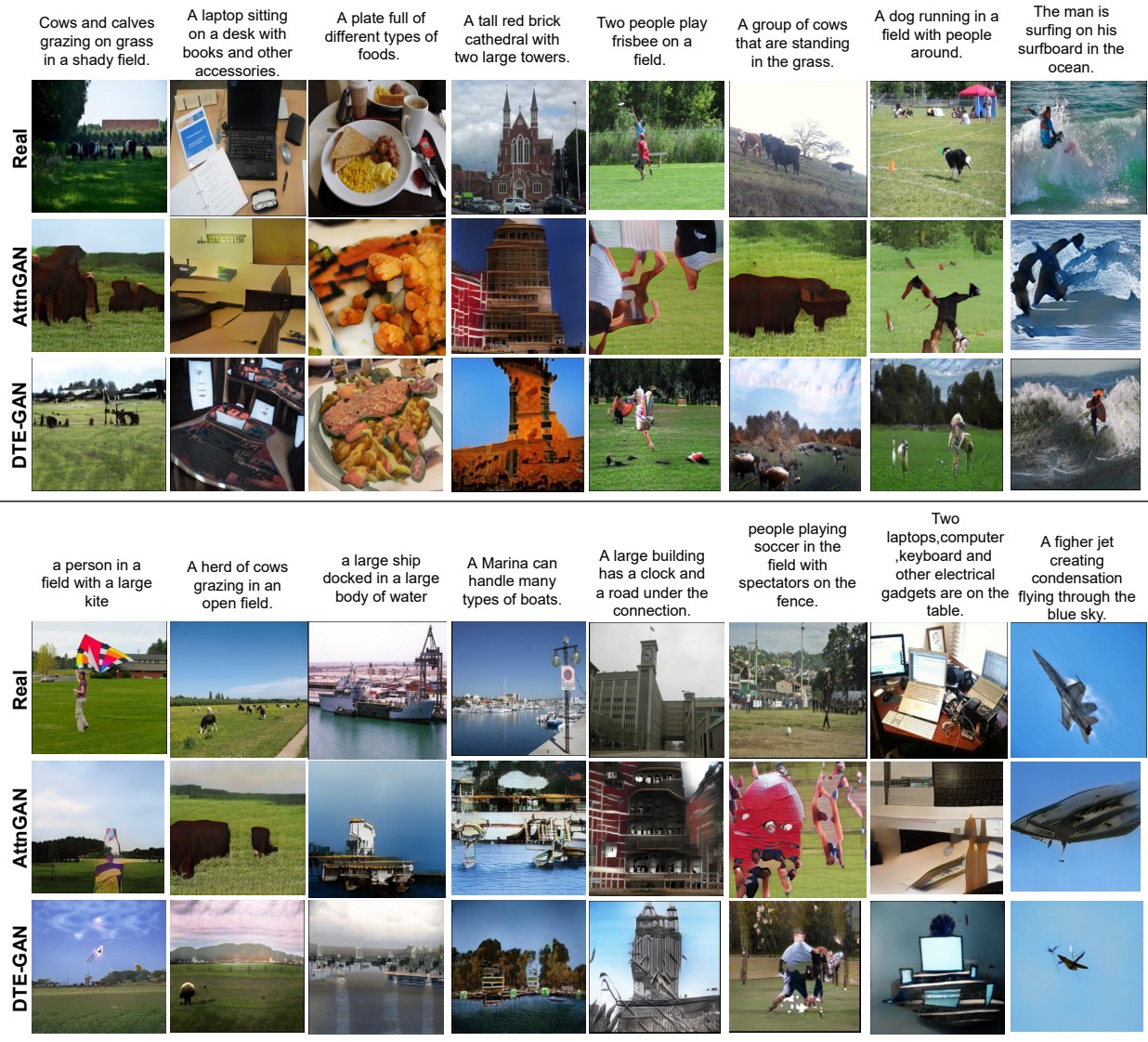

Figure 6: Images generated by DTE-GAN (ours) on COCO dataset and compared with AttnGAN Xu et al. (2018).

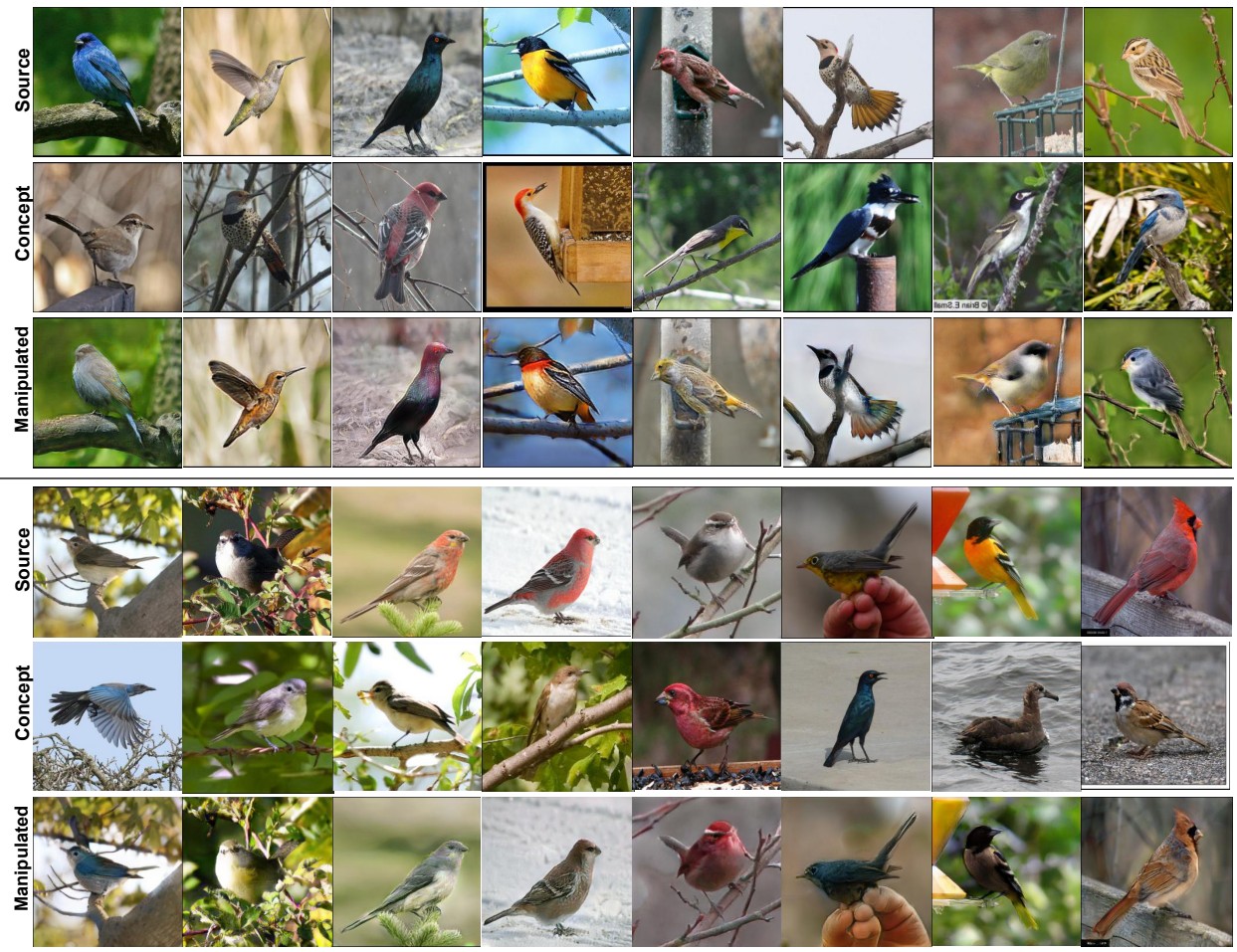

Figure 7: Examples of manipulated images generated by LightWeight GAN Li et al. (2020) using DTE-GAN pre-trained embeddings on CUB dataset. Source images (row 1) are manipulated by the text from concept images (row 2) and results in manipulated images (row 3).

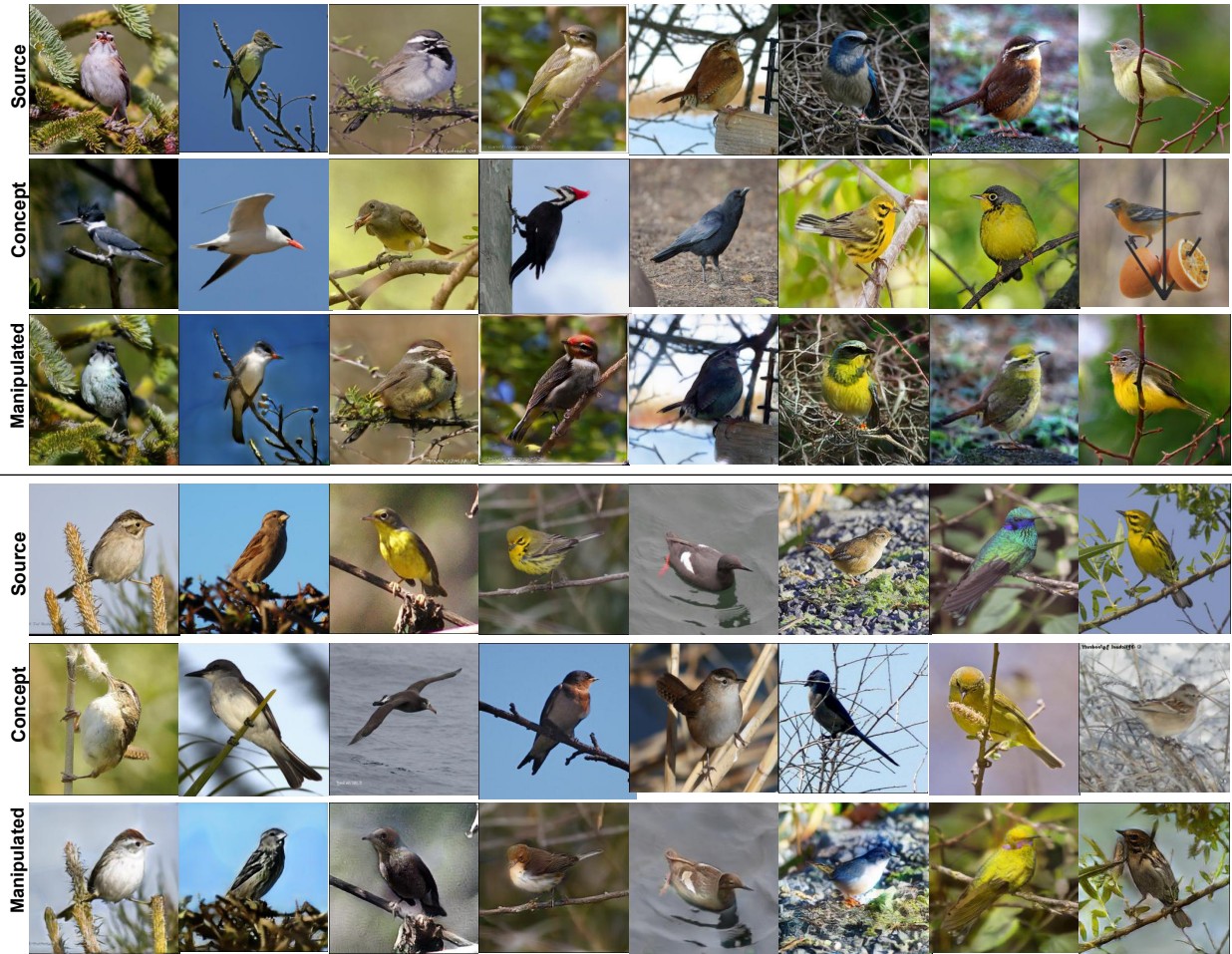

Figure 8: Examples of manipulated images generated by Light Weight GAN Li et al. (2020) using DTE-GAN pre-trained embeddings on CUB dataset. Source images (row 1) are manipulated by the text from concept images (row 2) and results in manipulated images (row 3).

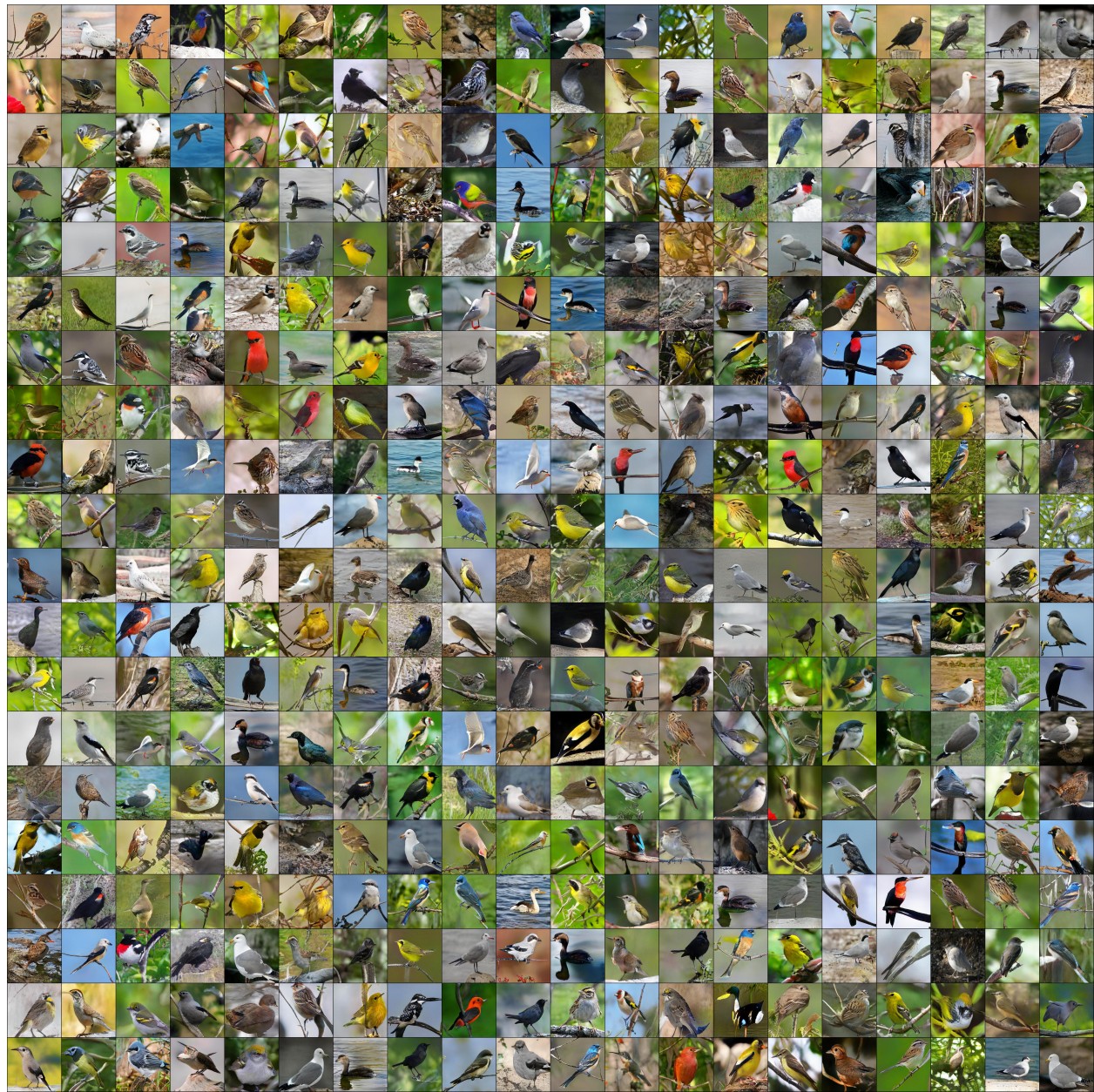

Figure 9: Random images generated on CUB dataset. (best viewed in full zoom and color)

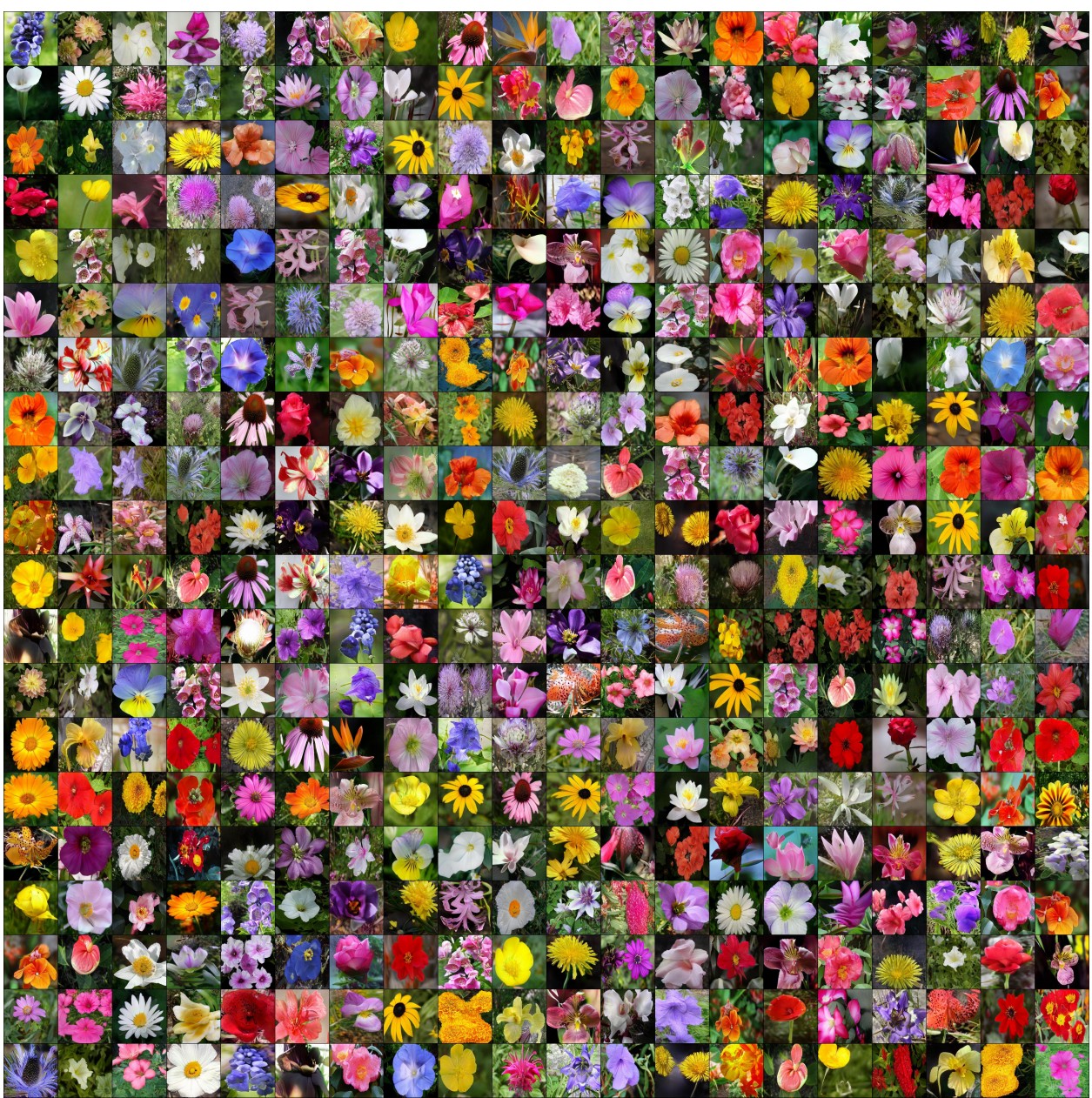

Figure 10: Random images generated on Oxford-102 dataset. (best viewed in full zoom and color)

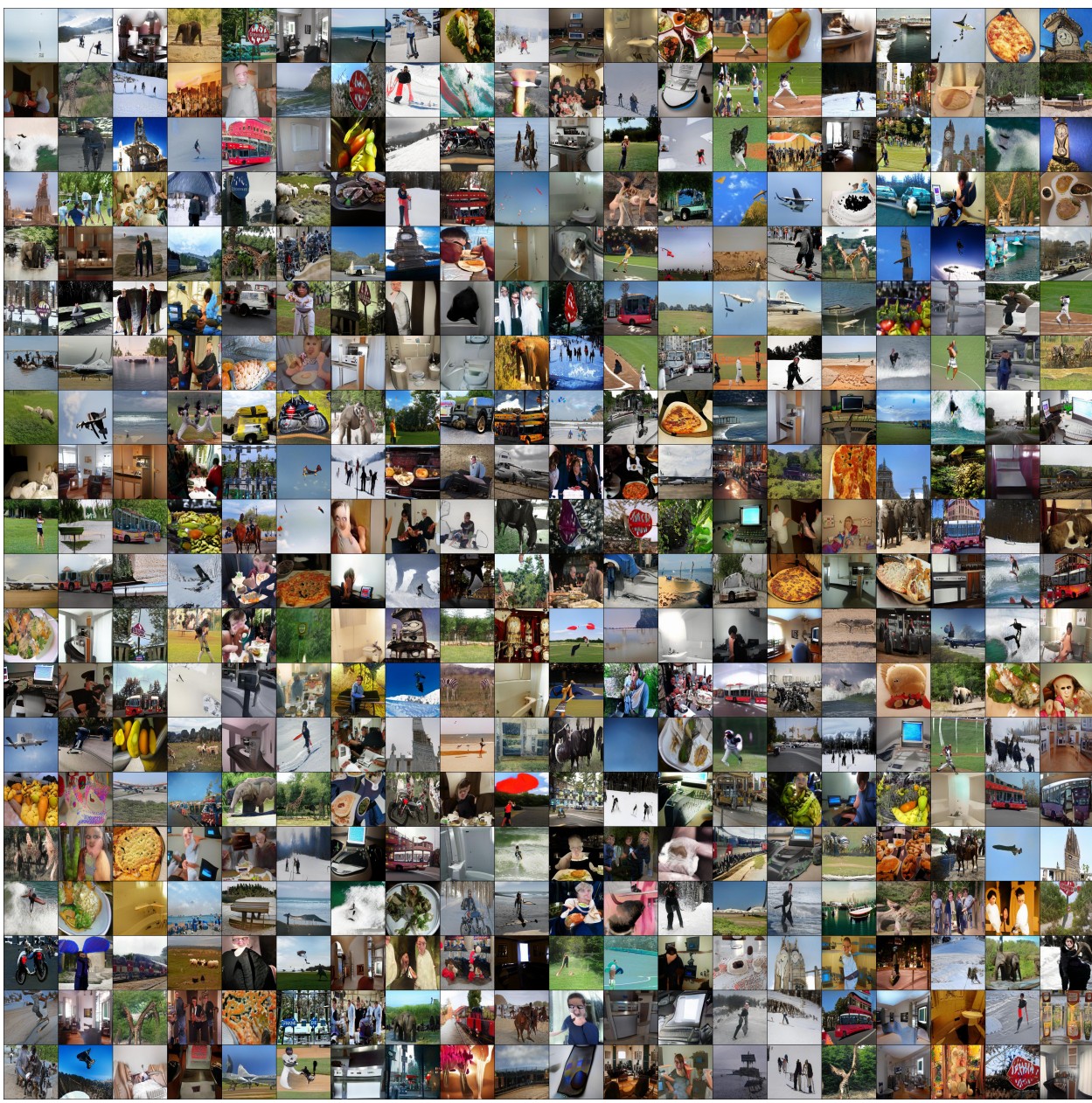

Figure 11: Random images generated on COCO dataset. (best viewed in full zoom and color)