# OpenReview forum: "End-to-end Training for Text-to-Image Synthesis using Dual-Text Embeddings"
_TMLR — Accepted by TMLR_

### Review · Reviewer_Wz7H · 2025-03-02

**Summary Of Contributions:**

Authors explored an approach to learning text embeddings specifically tailored to the T2I synthesis network, trained in an end-to-end fashion. Their dual text embedding approach combining both generative and discriminative embeddings (Fig1).

**Audience:**

Yes

**Broader Impact Concerns:**

no concern at the moment

**Claims And Evidence:**

Yes

**Requested Changes:**

What is the resolution of generated images? e.g. COCO. I highly recommend the authors to include comparisons to latest T2I generations such as Stable Diffusion, given that it is open sourced.
Even more stronger and more impactful is the application of the proposed method on recent diffusion models."This method of focusing solely on generator-side embeddings can be seamlessly incorporated into current diffusion-based Text-to-Image models"

**Strengths And Weaknesses:**

The evaluation section should be improved. The comparison to SOTA models actually does not contain SOTA T2I models, such as stable diffusion. A good work on this domain should include comparison to latest diffusion models too, but this work only covers more GAN works.

---

> ### Author Response · Authors · 2025-05-07
> **Response to Reviewer Wz7H**
>
> We thank the reviewer $\textbf{Wz7H}$ for the helpful feedback and for recognizing our contribution in learning task-specific text embeddings for text-to-image generation. Your suggestions have helped us strengthen the paper further.
>
> $$
> \textbf{Q1. What is the resolution of generated images.}
> $$
>
>
> $\textbf{A1.}$ We generate images at a resolution of 256×256, following the setup used in lightweight GAN-based models [1,2,3]. Our comparisons are made with similar GAN baselines to ensure a fair and consistent evaluation.
>
>
> $\textbf{Q2. Include comparison to recent large scale models.}$
>
> $\textbf{A2.}$ We have added a comparison to recent Text-to-Image synthesis approaches on the COCO dataset in Table 10. This includes models based on GANs, diffusion, autoregressive methods, and hybrid models that combine diffusion and GAN losses. Our DTE embeddings can be directly incorporated into such models. Future work could explore the integration of learnable text embeddings within diffusion and autoregressive architectures to further improve alignment and generation quality.
>
>
> $$
> \textbf{References :}
> $$
>
> [1] Xu, Tao, et al. "Attngan: Fine-grained text to image generation with attentional generative adversarial networks." Proceedings of the IEEE conference on computer vision and pattern recognition. 2018.
>
> [2] Tao, Ming, et al. "Df-gan: A simple and effective baseline for text-to-image synthesis." Proceedings of the IEEE/CVF conference on computer vision and pattern recognition. 2022.
>
> [3] Liao, Wentong, et al. "Text to image generation with semantic-spatial aware gan." Proceedings of the IEEE/CVF conference on computer vision and pattern recognition. 2022.

---

### Review · Reviewer_FnYf · 2025-03-24

**Summary Of Contributions:**

This paper proposes Dual Text Embeddings (DTE) to enhance the image quality and text-to-image alignment in text-to-image (T2I) models. Unlike previous works, the proposed DTE-GAN method breaks apart from using pre-trained, generic embeddings by training two distinct text embeddings within the neural network: one optimized for improving image realism and the other for better text-to-image alignment, which is integrated into the GAN architecture via adversarial and multimodal contrastive losses. The model is evaluated on three benchmark datasets: Oxford-102, Caltech-UCSD, and MS-COCO, demonstrating superior or competitive performance in terms of Inception Score, FID, and R-precision.

**Audience:**

Yes

**Broader Impact Concerns:**

Currently, there are no broader impact concerns.

**Claims And Evidence:**

Yes

**Requested Changes:**

It would strengthen the paper and benefit the readers if the red, green, blue colors used in the tables are substituted with bold, underline, etc.

**Strengths And Weaknesses:**

### Strengths

A notable strength of this work is the explicit separation of embeddings into two distinct roles: one for image quality and the other for semantic alignment. This separation promotes each embedding to specialize in its respective objective, leading to improved performance across standard benchmarks. The end-to-end training of these embeddings within the synthesis framework avoids reliance on fixed, pre-trained models.


### Weaknesses

While the current experimental results suggest that the dual embedding approach contributes to disentangling the roles of image quality and text-image alignment, this claim is currently supported indirectly through ablation studies. The paper would benefit from additional experiments that more explicitly analyze or visualize the distinct functionalities of each embedding. For instance, perturbation-based analyses (such as injecting controlled noise into each embedding) or latent space walking could offer deeper insights into their respective contributions.

---

> ### Author Response · Authors · 2025-05-07
> **Response to Reviewer FnYf**
>
> We thank Reviewer $\textbf{FnYf}$ for the helpful and encouraging feedback. We appreciate your recognition of our dual embedding approach and the end-to-end training design. Your suggestions helped us improve the paper further.
>
> $$
> \textbf{Q1. Visual results for latent space walking.}
> $$
> $\textbf{A1.}$ We have added visual results in Appendix Section C showing sentence interpolation [1]. The results include images generated using interpolated embeddings from the discriminator-side only, generator-side only, and both embeddings together. This provides a clearer visual understanding of how each embedding contributes to the generation process.
>
> $$
> \textbf{Q2.  red, green, blue colors used in the tables are substituted with bold, underline}
> $$
> $\textbf{A2.}$  We have made the changes as suggested by the reviewer by replacing red, green, and blue colors in the tables with formatting styles such as bold and underline. We thank the reviewer for the helpful suggestion.
>
> $$
> \textbf{Reference : }
> $$
>
> [1] Reed, Scott, et al. "Generative adversarial text to image synthesis." International conference on machine learning. PMLR, 2016.

---

### Review · Reviewer_BMR5 · 2025-04-09

**Summary Of Contributions:**

The authors propose DTE-GAN, a GAN-based framework. It is for text-to-image synthesis where the text embeddings are trained along with the image generator in an end-to-end manner. Furthermore, DTE-GAN decouples the learning of text embeddings in a dual framework that optimizes photorealism and text-image alignment.

**Audience:**

Yes

**Broader Impact Concerns:**

No ethical concern.

**Claims And Evidence:**

Yes

**Requested Changes:**

* Could the authors ablate the design choice of the Bi-LSTM-based text encoder used in DTEGAN?
* What captions were used in Figure 6?
* What is $\lambda_{3}$ in eqn. 12?
* How does the loss combination ensure learning of photorealism and text-image alignment? What is the intuition behind it?
* The benchmark datasets CalTech and Oxford don't have captions for the images. How were they generated?

**Strengths And Weaknesses:**

## Strengths

* The framework is well ablated and has a motivation. It also paves a way forward for the researchers to think about whether text embeddings should be trained/optimized for text-to-image synthesis.
* It uses simple text encoders, discarding the computationally costly ones.
* It shows the merits of training an image generator along with a decoupled text embedding learning strategy.

## Weaknesses

* The paper mentions the use of CLIP in modern diffusion-based T2I models. However, that is not true. Many models like PixArt-Alpha [1], SANA [2], etc., don't use it and instead rely on T5 [3] and Gemma [4] for text embedding. Imagen [5] even suggests scaling up the size of text encoders for more photorealistic generation.
* It doesn't use any metrics that are known to assess text-image alignment and image manipulation (ClipScore [6] and directional similarity [7], respectively).
* It is unclear what training dataset was used for DTE-GAN.

## References

[1] PixArt-α: Fast Training of Diffusion Transformer for Photorealistic Text-to-Image Synthesis; Chen et al.; 2023; https://arxiv.org/abs/2310.00426

[2] SANA: Efficient High-Resolution Image Synthesis with Linear Diffusion Transformers; Xie et al.; 2024; https://arxiv.org/abs/2410.10629

[3] Exploring the Limits of Transfer Learning with a Unified Text-to-Text Transformer; Raffel et al.; 2019; https://arxiv.org/abs/1910.10683

[4] Gemma: Open Models Based on Gemini Research and Technology; Gemma Team; 2024; https://arxiv.org/abs/2403.08295

[5] Photorealistic Text-to-Image Diffusion Models with Deep Language Understanding; Saharia et al.; 2022; https://arxiv.org/abs/2205.11487

[6] CLIPScore: A Reference-free Evaluation Metric for Image Captioning; Hessel et al.; 2021; https://arxiv.org/abs/2104.08718

[7] InstructPix2Pix: Learning to Follow Image Editing Instructions; Brooks et al.; 2022; https://arxiv.org/abs/2211.09800

---

> ### Author Response · Authors · 2025-05-07
> **Response to Reviewer BMR5 (Part-1)**
>
> We thank Reviewer $\textbf{BMR5}$ for the thoughtful review and useful comments. We appreciate your positive feedback on our motivation and framework design. Your suggestions helped us improve the paper further. Please find our responses below.
>
> $$
> \textbf{Q1. The paper mentions the use of CLIP in modern diffusion-based T2I models. However, that is not true. Many models like PixArt-Alpha [1], SANA [2], etc., don't use it and instead rely on T5 [3] and Gemma [4] for text embedding. Imagen [5] even suggests scaling up the size of text encoders for more photorealistic generation.}
> $$
>
> $\textbf{A1.}$ Thank you for the clarification. We agree that many recent models such as PixArt-Alpha, SANA, and Imagen use encoders like T5 and Gemma instead of CLIP[6]. At the same time, CLIP is still widely used in current state-of-the-art models like OpenAI's DALL·E 3 [7] and UfoGen [8]. Some recent works, such as Stability AI’s Stable Diffusion 3 [9], also explore combining CLIP and T5 in the same framework.
>
> Unlike these methods, which rely on large pre-trained text encoders, our approach jointly learns the text encoder and image generator from scratch. Our decoupled training setup focuses on learning text representations that are optimized specifically for the image generation task.
>
> $$
> \textbf{Q2.  Could the authors ablate the design choice of the Bi-LSTM-based text encoder used in DTEGAN?}
> $$
>
> $\textbf{A2.}$ We have included an ablation study on text encoder choices in the appendix under "Text Encoding Scheme." Bi-LSTM was chosen to ensure a fair comparison with DAMSM embeddings [10], which are standard in lightweight GAN setups on COCO and CUB datasets. Our goal is to highlight the effectiveness of the proposed decoupled learning framework rather than relying on large, computationally intensive encoders for improved performance.
>
> $$
> \textbf{Q3. What captions were used in Figure 6?}
> $$
> $\textbf{A3.}$ We have added the corresponding captions to Figure 6. The captions are related to concept images.
>
> $$
> \textbf{Q4. What is λ3  in eqn. 12?}
> $$
> $\textbf{A4.}$ λ₃ is a hyperparameter that we set to 1 in all our experiments. It controls the contribution of the text-image alignment loss within the overall loss function. Adjusting this value can help balance the importance of alignment versus photorealism during training.
>
> $$
> \textbf{Q5.  The benchmark datasets CalTech and Oxford don't have captions for the images. How were they generated?}
> $$
> $\textbf{A5.}$ We used manually created image captions for the CUB and Oxford-102 datasets, which are publicly available [11]. These captions are commonly used in text-to-image synthesis tasks.
>
> $$
> \textbf{Q6.  Doesn't use any metrics that are known to assess text-image alignment and image manipulation. }
> $$
>
> $\textbf{A6.}$ For all our experiments, we use R-precision based on DAMSM embeddings, fine-tuned to the CUB and COCO datasets, to measure text-image alignment, as proposed in AttnGAN [10]. This metric is commonly used in lightweight GAN setups for these datasets. Additionally, we include Manipulative Precision [12] scores to assess performance on the image manipulation task.

---

> > ### Author Response · Authors · 2025-05-07
> > **Response to Reviewer BMR5 (Part-2)**
> >
> > $$\textbf{Q7. How does the loss combination ensure learning of photorealism and text-image alignment? What is the intuition behind it?}
> > $$
> > $\textbf{A7.}$ The discriminator-side embeddings are trained using only the text-image alignment loss on real image-caption pairs, similar to DAMSM and CLIP embeddings. This helps capture the semantic alignment between text and images. On the other hand, the generator-side embeddings are learned jointly with the generator and contribute directly to improving image generation quality through the generator’s loss. The design of the decoupled embeddings and the effect of the associated losses are ablated in Subsections 4.3.4, 4.3.5, and 4.3.6.
> >
> > $$
> > \textbf{References :}
> > $$
> >
> > [1] PixArt-α: Fast Training of Diffusion Transformer for Photorealistic Text-to-Image Synthesis; Chen et al.; 2023; https://arxiv.org/abs/2310.00426
> >
> > [2] SANA: Efficient High-Resolution Image Synthesis with Linear Diffusion Transformers; Xie et al.; 2024; https://arxiv.org/abs/2410.10629
> >
> > [3] Exploring the Limits of Transfer Learning with a Unified Text-to-Text Transformer; Raffel et al.; 2019; https://arxiv.org/abs/1910.10683
> >
> > [4] Gemma: Open Models Based on Gemini Research and Technology; Gemma Team; 2024; https://arxiv.org/abs/2403.08295
> >
> > [5] Photorealistic Text-to-Image Diffusion Models with Deep Language Understanding; Saharia et al.; 2022; https://arxiv.org/abs/2205.11487
> >
> > [6] Radford, Alec, et al. "Learning transferable visual models from natural language supervision." International conference on machine learning. PmLR, 2021.
> >
> > [7] Betker, James, et al. "Improving image generation with better captions." Computer Science. https://cdn. openai. com/papers/dall-e-3. pdf 2.3 (2023): 8.
> >
> > [8] Xu, Yanwu, et al. "Ufogen: You forward once large scale text-to-image generation via diffusion gans." Proceedings of the IEEE/CVF Conference on Computer Vision and Pattern Recognition. 2024.
> >
> > [9] Esser, Patrick, et al. "Scaling rectified flow transformers for high-resolution image synthesis, 2024." URL https://arxiv. org/abs/2403.03206 2.
> >
> > [10] Xu, Tao, et al. "Attngan: Fine-grained text to image generation with attentional generative adversarial networks." Proceedings of the IEEE conference on computer vision and pattern recognition. 2018.
> >
> > [11] Reed, Scott, et al. "Learning deep representations of fine-grained visual descriptions." Proceedings of the IEEE conference on computer vision and pattern recognition. 2016.
> >
> > [12] Li, Bowen, et al. "Manigan: Text-guided image manipulation." Proceedings of the IEEE/CVF conference on computer vision and pattern recognition. 2020.

---

### Decision · Action_Editor_r8Qm · 2025-05-20

**Recommendation:** Accept as is

**Comment:**

The reviewers all acknowledge the interest and value of separating the role of the text embeddings in text-to-image generation. They initially expressed a few concerns regarding the empirical evaluation of the work. These concerns, however, were cleared by the author's responses and revisions of the paper. As such, all reviewers recommended acceptance in their final recommendation.

**Audience:**

All the reviewer agree that there is a TMLR audience for this work.

**Claims And Evidence:**

All the reviewers acknowledge that the claims are supported by sufficient evidence.